# Dopaminergic regulation of vestibulo-cerebellar circuits through unipolar brush cells

Jose Ernesto Canton-Josh, Joanna Qin, Joseph Salvo, Yevgenia Kozorovitskiy*

Department of Neurobiology, Northwestern University, Evanston, United States

**Summary** While multiple monoamines modulate cerebellar output, the mechanistic details of dopaminergic signaling in the cerebellum remain poorly understood. We show that dopamine type 1 receptors (Drd1) are expressed in unipolar brush cells (UBCs) of the mouse cerebellar vermis. Drd1 activation increases UBC firing rate and post-synaptic NMDAR -mediated currents. Using anatomical tracing and in situ hybridization, we test three hypotheses about the source of cerebellar dopamine. We exclude midbrain dopaminergic nuclei and tyrosine hydroxylase-positive Purkinje (Pkj) cells as potential sources, supporting the possibility of dopaminergic co-release from locus coeruleus (LC) axons. Using an optical dopamine sensor GRAB$_{DA2h}$, electrical stimulation, and optogenetic activation of LC fibers in the acute slice, we find evidence for monoamine release onto Drd1-expressing UBCs. Altogether, we propose that the LC regulates cerebellar cortex activity by co-releasing dopamine onto UBCs to modulate their response to cerebellar inputs. Pkj cells directly inhibit these Drd1-positive UBCs, forming a dopamine-sensitive recurrent vestibulo-cerebellar circuit.

## Editor's evaluation

This paper uses an impressive battery of anatomical and functional approaches to establish that a subpopulation of unipolar brush cells (UBCs) in the vestibulocerebellum express Dr1a dopamine receptors and that these cells are a target of locus-coeruleus-mediated dopamine release. It also demonstrates a recurrent microcircuit between UBCs and Purkinje cells, and highlights possible consequences of dopaminergic modulation of this microcircuit on cerebellar output. It will be interesting to see future in vivo studies examine the functional impact of the findings described here, which extend our understanding of neuromodulation in the cerebellum.

*For correspondence:
Yevgenia.Kozorovitskiy@
northwestern.edu

**Competing interest:** The authors declare that no competing interests exist.

## Introduction

Cerebellar activity plays a critical function in fine motor learning (*Kalmbach et al., 2010*; *Raymond and Lisberger, 1998*; *Woodruff-pak, 1993*). The cerebellum is defined by its clearly organized cytoarchitecture; however, important anatomical, molecular, and functional differences between lobules are beginning to emerge. These differences include specialized neuronal classes present only in certain regions of the cerebellar cortex. Electrophysiological and behavioral data indicate that these varying motifs are likely relevant for the function of specific lobules. One key difference between cerebellar lobules is the distribution of unipolar brush cells (UBCs), glutamatergic interneurons enriched in the granular layer of vestibulo-cerebellum, which is known to be involved in processing vestibular sensory input (*Mugnaini and Floris, 1994*). UBCs reside in the granular layer and receive inputs from mossy fibers. Each UBC axon diverges, releasing glutamate onto multiple local granule cells or other UBCs. Since UBCs are recurrently connected, they are poised to amplify and temporally extend the influence of excitatory inputs into the cerebellum (*Requarth and Sawtell, 2014*). Alternatively, UBCs—receiving

a single mossy fiber input—could form a 'labeled line' input to the granular layer, helping distinguish between inputs from semicircular canals and vestibular nuclei (*Balmer and Trussell, 2019*). The relatively recent discovery of UBCs in the late 1970s (*Altman and Bayer, 1977*) and characterization in the 1990s (*Mugnaini and Floris, 1994*), their sparseness in the cerebellum, and the lack of tools to isolate them have limited our understanding of their functional connectivity.

Differences in inputs and expression patterns of neuromodulatory receptors also distinguish cerebellar regions (*Cerminara et al., 2015*). While it is clear that norepinephrine (NE) regulates cerebellar output, the mechanistic details of neuromodulatory effects for other amines remain poorly understood (*Basile and Dunwiddie, 1984*; *Carey and Regehr, 2009*; *Lanore et al., 2019*; *Lippiello et al., 2015*). Previous reports posit the expression of dopamine type 1 receptor (Drd1) within the granular layer of the cerebellum (*Locke et al., 2018*; *Panagopoulos et al., 1993*). Additionally, several groups have previously reported dopaminergic projections from the substantia nigra pars compacta (SNc) and the ventral tegmental area (VTA), targeting the cerebellar cortex of rats and monkeys (*Melchitzky and Lewis, 2000*; *Panagopoulos et al., 1991*). Yet, recent work in mice did not find evidence for projections from midbrain dopaminergic nuclei to the cerebellum. Another potential source of dopamine to the cerebellum is from a subset of tyrosine hydroxylase (TH) positive Purkinje (Pkj) cells. Vestibulocerebellar Pkj cells express TH, the rate-limiting enzyme in the classical dopamine (DA) synthesis pathway (*Abbott et al., 1996*; *Huang et al., 2016*; *Sawada et al., 2004*; *Takada et al., 1993*). One prior report details a dendritic release of dopamine by Pkj cells onto their own dendrites, expressing dopamine type 2 receptor (Drd2) (*Kim et al., 2009*). It is possible that these Pkj cells could also be the source of dopamine for local UBCs expressing Drd1 receptors. A third potential source of cerebellar dopamine is the locus coeruleus (LC), recently demonstrated to release dopamine as well as NE in the hippocampus (*Kawahara et al., 2001*; *Kempadoo et al., 2016*; *Takeuchi et al., 2016*). The proposed mechanism of action for this dual amine co-release is incomplete conversion of dopamine to NE in the vesicles. If incomplete conversion also takes place in the cerebellum, then dopamine release could enhance activity levels in the UBC network or coordinate the timing of their activity.

Altogether, previous research is consistent with the possibility of cerebellar dopamine signaling but many unknowns remain. Here, using a combination of anatomical characterization, electrophysiology, two-photon laser scanning microscopy, and glutamate photolysis, along with optogenetics and dopamine sensor imaging, we resolve this uncertainty by characterizing the expression of Drd1 receptors in the cerebellar cortex, defining the role of these receptors and the function of Drd1-positive UBCs within cerebellar circuits, and revealing the source of cerebellar dopamine.

## Results

### Expression of *Drd1a* transcripts in UBCs of lobules IX/X

Analysis of Drd1-Cre;tdTomato (tdT) expression in the cerebellar cortex revealed numerous small tdT+ cells distributed across the cerebellar granular layer (*Figure 1A*), enriched in lobules IX and X of the cerebellar vermis, lobules VIb and VII, and the paraflocculus. These regions of the cerebellum are known to contain many UBCs. Confocal imaging confirmed that cerebellar tdT+ cells possess the morphological features of UBCs: a single large dendritic brush and a medium-sized soma relative to granule cells (*Figure 1B*). All tdT+ cells in the cerebellum were positive for nuclear protein Eomesodermin (Tbr2), a known marker for UBCs (*Englund et al., 2006*). However, only 35% of Tbr2+ UBCs expressed tdT, suggesting that a subpopulation of UBCs expresses tdT driven by Drd1 promoter activity (*Figure 1C* and *Figure 1—figure supplement 1*). Since two primary classes of UBCs have been previously described (*Kim et al., 2012*; *Zampini et al., 2016*), we relied on known immunohistochemistry markers to characterize tdT+ UBCs. The majority of tdT+ neurons expressed metabotropic glutamate receptor type 1 (mGluR1) (*Figure 1D*) (71%, n = 2 mice, 779 cells). The mGluR1+ and tdT+ populations did not overlap completely, as only 65% of mGluR1+ UBCs expressed tdT. Very few tdT+ UBCs belonged to calretinin+ (CR) type: 5.3% of tdT+ UBCs expressed CR, and 12.5% of CR+ UBCs expressed tdT (*Figure 1E*). Given these data, we infer that in the Drd1-Cre;tdT cross, all cerebellar tdT+ cells are UBCs, and that these tdT+ UBCs are a subset of the total UBC population, mostly comprised of mGluR1+ ON UBCs. A prior report using a different Drd1-Cre mouse line reported the expression of Drd1 receptors in deep cerebellar nuclei (*Locke et al., 2018*). In contrast, we do not see expression of tdT in the cerebellar nuclei in this transgenic cross. Because Cre recombinase lines can

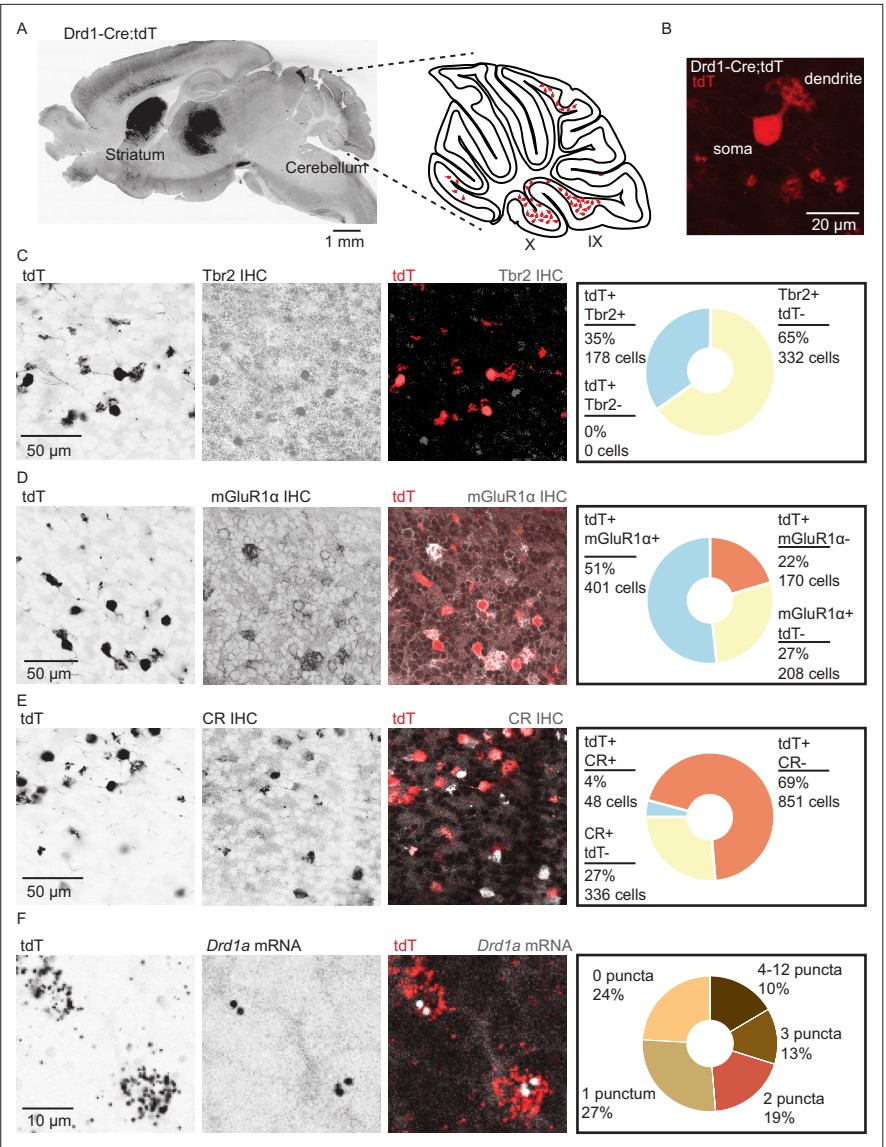

**Figure 1.** Dopamine type 1a receptor transcripts are expressed in mGluR1+ unipolar brush cells. (**A**) Left: A whole brain parasagittal section from a Drd1-Cre;tdTomato (tdT) reporter cross. Right: A cartoon view of a parasagittal slice of the cerebellar vermis. tdT expressing unipolar brush cells (UBCs) are abundant in lobules IX/X of the vermis. Scale bar, 1 mm. (**B**) A confocal image of a tdT+ UBC in the cerebellum of a Drd1-Cre mouse, showing the characteristic dendritic brush morphology of a UBC. Scale bar, 20 μm. (**C**) Left: Confocal images of immunofluorescent labeling of Eomesodermin (Tbr2) in lobules IX/X of the cerebellar vermis. Right: Pie chart describing overlap of cerebellar tdT+ cells and Tbr2. All tdT+ cells express Tbr2, and a subpopulation (~35%) of Tbr2 cells expresses tdT (n = 2 mice, 510 cells). Inverted grayscale, single-channel images; in merged images, red represents tdT signal, gray represents antibody labeling. Scale bar, 50 μm. (**D**) Left: Similar to C, but for metabotropic glutamate receptor type 1 (mGluR1). Right: Out of all tdT+ UBCs, 70% expressed mGluR1; 66% of mGluR1+ UBCs were positive for tdT (n = 2 mice, 779 cells). Scale bar, 50 μm. (**E**) Left: Similar to C–D, but for calretinin (CR). Right: little overlap was observed between CR and tdT+ UBCs: 5.3% of tdT+ UBCs express CR, and 12.5% of CR+ express tdT (n = 2 mice, 1235 cells). Scale bar, 50 μm. (**F**) Left: Sample confocal images of fluorescent in situ hybridization, labeling tdT, and *Drd1a* transcripts in cerebellar UBCs. Inverted grayscale lookup table (LUT) is used for single-channel images. In merged images, red represents tdT signal, gray represents Drd1 labeling. Right: The majority (~76%) of tdT+ UBCs express dopamine type 1a receptor (*Drd1a*) transcripts (n = 4 mice, 2553 cells). Scale bar, 10 μm.

The online version of this article includes the following source data and figure supplement(s) for figure 1:

**Source data 1.** Numerical data for graphs in *Figure 1*.

*Figure 1 continued on next page*

Figure 1 continued

**Figure supplement 1.** Co-localization of tdTomato expressing unipolar brush cells with unipolar brush cell subtype markers and dopamine type 1a receptor mRNA.

**Figure supplement 1—source data 1.** Numerical data for graphs in *Figure 1—figure supplement 1*.

show developmentally regulated expression patterns, next we evaluated whether tdT+ UBCs contain Drd1 receptors using fluorescent in situ hybridization (FISH) to label *tdT* and *Drd1a* mRNA transcripts. High density *tdT* transcripts were used to create ROIs in ImageJ (*Schindelin et al., 2012*; *Schneider et al., 2012*) to examine co-localization with *Drd1a* transcripts. We found that 76% of *tdT+* (*Figure 1F*) UBCs expressed *Drd1a* transcripts in low abundance (n = 4 animals, 2553 cells) across an age range from P20s to P60s (*Figure 1—figure supplement 1B, C*). The density of *Drd1a* puncta was 40-fold enriched in the volume of *tdT+* ROIs, compared with regions outside the cell bodies (*Figure 1—figure supplement 1D*). These data suggest that tdT+ UBCs express modest levels of Drd1 receptors.

## Functional characterization of Drd1 activation in UBCs

To evaluate the consequences of Drd1 receptor activation, we used current clamp recordings of tdT+ UBCs within lobules IX/X of the cerebellar vermis (*Figure 2A–F and Figure 1—figure supplement 1A-F*). Puff application of selective Drd1 agonist SKF81297 increased the firing rate of tdT+ UBCs over a period of 10 min (*Figure 2C–D and Figure 1—figure supplement 1A*) (2.48 Hz increase, 110% increase from baseline, n = 13 cells, paired Wilcoxon signed rank test, p=0.02). We recorded two separate control datasets, one where we applied SKF81297 in the presence of a Drd1 antagonist cocktail (SKF83566, SCH39166, and SCH23390; n = 13 cells, paired Wilcoxon signed rank test, p=0.49), and an independent dataset where we delivered puff application of artificial cerebrospinal fluid (ACSF; n = 13 cells, paired Wilcoxon signed rank test, p=0.33) (*Figure 2E, F*). We compared the changes in firing rate, resting membrane potential, and relative input resistance between the baseline period and SKF application. Increases in firing rate, membrane potential, and relative input resistance were detected in response to Drd1 activation, but absent in both control groups (*Figure 2E, F*). Thus, UBCs respond to Drd1 activation with an enhancement in activity, consistent with previous experiments in the striatum and prefrontal cortex (*Aosaki et al., 1998*; *Chen et al., 2007*; *Hernández-López et al., 1997*).

Prior research shows an increase in N-methyl-D-aspartate receptor (NMDAR) currents during the activation of Drd1 receptors on neurons in the prefrontal cortex (*Chen et al., 2004*). Using electrical stimulation of mossy fibers, we found an increase in peak amplitude of NMDAR currents after the flow in of 10 µM SKF81297 (*Figure 2I–K*) (15% increase in peak current, n = 9 cells, paired t-test, p=0.0251). These experiments were done in the presence of blockers for excitatory AMPA receptor currents and inhibitory neurotransmission: 1 µM NBQX, 1 µM gabazine, and 1 µM strychnine. In a subset of experiments, we verified that the measured currents were blocked by the application of NMDAR antagonist CPP (10 µM, n = 5 cells) (*Figure 2—figure supplement 1G*). Since UBCs are recurrently connected, the increase in NMDAR currents may be downstream of a post-synaptic mechanism or a consequence of increased excitability in presynaptic UBCs. To distinguish between post-synaptic and pre-synaptic mechanisms, we used two-photon focal uncaging of MNI L-glutamate to bypass pre-synaptic terminals (*Figure 2L–M*; *Kozorovitskiy et al., 2015*; *Xiao et al., 2018*). In the presence of 1 mM MNI L-glutamate for uncaging, 1 µM tetrodotoxin (TTX) to silence spontaneous release, and drugs to isolate NMDAR currents, we uncaged glutamate near the dendritic brush of voltage clamped UBCs held at +40 mV. NMDAR-mediated currents in response to 1-ms long pulses of 725 nm laser light were compared for experiments with and without 10 µM SKF81297 present in the bath. Current amplitudes increased with Drd1 activation, consistent with a post-synaptic mechanism of action (*Figure 2N–O*) (51% increase of average amplitude, n = 13 cells, unpaired t-test, p=0.0068). These experiments support an NMDAR-mediated component to Drd1 activation in UBCs, although given prior findings in the striatum and prefrontal cortex, it is likely that additional mechanisms account for observed changes in excitability (*Gorelova and Yang, 2000*; *Surmeier and Kitai, 1993*).

## Defining the source of cerebellar dopamine

Three TH+ populations are poised to serve as the source of dopamine to the cerebellum based on prior data: the midbrain dopaminergic cells of the SNc and the VTA, local TH+ Pkj cells of the cerebellum (*Takada et al., 1993*), and the nearby LC neurons that release dopamine as well as NE in other

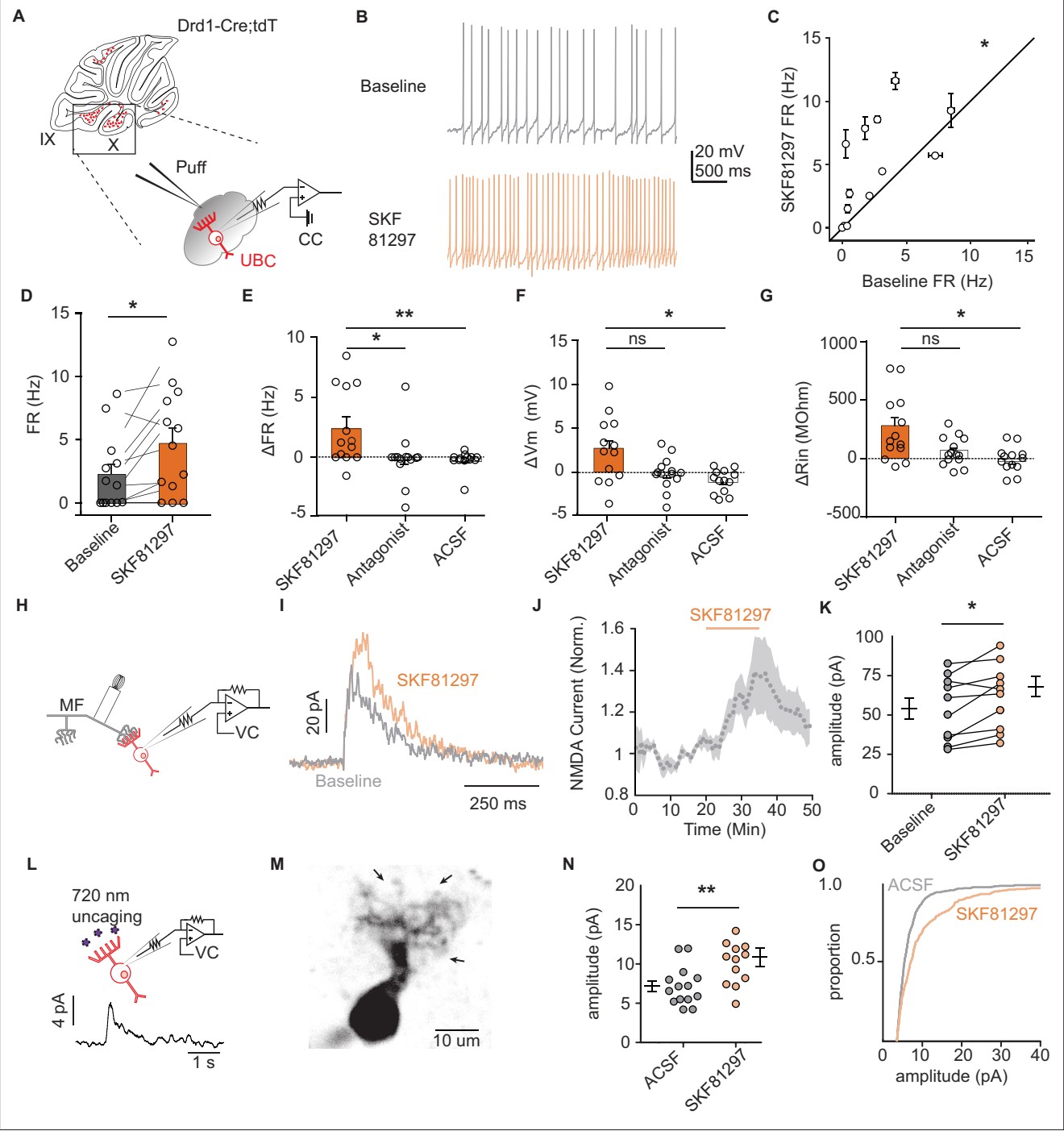

**Figure 2.** Dopamine type 1 receptor activation increases the firing rate and NMDAR currents in unipolar brush cells. (**A**) Diagram of experimental setup with tdTomato (tdT+) unipolar brush cells (UBCs) in whole-cell current clamp configuration. FR was measured during baseline period, followed by application of Drd1 agonist SKF81297 (500 µM) with 300-ms long puffs from a proximally located pipette. (**B**) Membrane voltage traces from an example cell which increased its firing rate (FR) in response to SKF81297 application. (**C, D**) Group comparison, mean FRs during the baseline period, and application of SKF81297 (n = 13 cells, paired Wilcoxon signed rank test, p=0.02, *p<0.05). (**E-G**) Changes in firing rate (ΔFR) in E, resting membrane potential (ΔmV) in F, and input resistance (ΔRin) in G after drug application. Comparisons are between three independent datasets. SKF81297: puff application of Drd1 agonist. Antagonist: SKF81297 was applied in the presence of Drd1 blockers 1 µM SKF83566, 1 µM SCH39166, and 1 µM SCH23390. Artificial cerebrospinal fluid (ACSF): puff application ACSF (Kruskal-Wallis test, FR p=0.0044, Vm p=0.0024, Rin p=0.02; Dunn's multiple comparison test, *p<0.05, **p<0.01). (**H**) Schematic of experimental setup. Voltage clamp recordings of tdT+ UBCs, paired with stimulation of mossy fiber inputs to evoke NMDA receptor mediated currents (intertrial interval [ITI], 5 s), in the presence of blockers for AMPA, GABA, and glycine receptors (1 µM NBQX, 1 µM gabazine, and 1 µM strychnine). (**I**) Sample traces show an increase in NMDAR current amplitudes during the application of SKF81297. (**J**) Trace

*Figure 2 continued on next page*

Figure 2 continued

of normalized peak amplitude of NMDAR currents across cells (n = 10 cells). Shaded region, ± SEM. (**K**) Summary plot of average evoked N-Methyl-D-aspartate receptor (NMDAR) current amplitudes before and after application of 10 µM SKF81297 (n = 10 cells, paired t-test, p=0.0251). (**L**) Top: Schematic of two-photon MNI-L-glutamate uncaging experiments. TdT+ UBCs were voltage clamped and imaged using a two-photon laser scanning microscope. Laser pulses (725 nm) directed near the dendritic brush (25 mW, 1 ms, 60 s ITI) were used to evoke NMDAR mediated currents. Bottom: Example current trace. (**M**) Two-photon imaging Z-projection of a tdT+ UBC. Arrows represent separate uncaging site locations. Scale bar, 10 µm. (**N**) NMDAR current amplitudes in ACSF vs 10 µM SKF81297 (unpaired t-test, p=0.0068, **p<0.01). Each data point is the average amplitude of NMDAR currents evoked by glutamate uncaging from one cell (ACSF n = 14 cells, SKF81297 n = 13 cells). (**O**) Cumulative distribution of uncaging-evoked NMDAR current peak amplitudes for ACSF (14 cells, 466 individual trials) and SKF81297 (13 cells, 349 individual trials) groups.

The online version of this article includes the following source data and figure supplement(s) for figure 2:

**Source data 1.** Numerical data for graphs in *Figure 2*.

**Figure supplement 1.** Characterization of dopamine type 1 receptor activation effects on unipolar brush cell firing and post-synaptic currents.

**Figure supplement 1—source data 1.** Numerical data for graphs in *Figure 2—figure supplement 1*.

brain regions (*Takeuchi et al., 2016*). First, to evaluate whether Drd1+ UBCs could receive inputs from midbrain dopamine neurons, we carried out retrograde tracing from lobules IX/X of the vestibulo-cerebellum in a Dat-Cre;tdT mouse cross. We found no support for the existence of this projection (*Figure 3A–C*), despite the abundance of Dat-Cre;tdT axonal fibers in the cerebellum (*Figure 3B*, inset). Retrograde labeling confirmed that these tdT+ axons originate from a pre-cerebellar brainstem nucleus, the lateral reticular nucleus (LRN), and not the SNc/VTA. LRN neurons are glutamatergic, forming mossy fiber rosettes within the cerebellum (*Rajakumar et al., 1992*; *Wu et al., 1999*). There are no prior reports of LRN neurons expressing proteins required for dopamine synthesis or packaging. To evaluate the expression of transcripts required for dopamine release, we carried out FISH labeling in the LRN and found that few neurons express mRNA for either TH or vesicular monoamine transporter type 2 (*Slc18a2/Vmat2*) (n = 3 mice, *Th*,19/135 LRN cells, *Slc18a2* 2/135 LRN cells) (*Figure 3—figure supplement 1B-D*).

Next, using FISH and antibody labeling, we confirmed previous reports of TH expression in Pkj cells of vermal lobules IX and X (*Figure 3D-F* and *Figure 3—figure supplement 1A*; *Austin et al., 1992*; *Sakai et al., 1995*; *Takada et al., 1993*). Despite the presence of *Th*, these neurons lacked the transcripts for dopa decarboxylase (*Ddc*), the enzyme responsible for converting L-DOPA to dopamine (*Figure 3E–F*) (n = 2 mice, 163 cells). Similarly, we found minimal expression of *Slc18a2* or dopamine active transporter (*Slc6a3/Dat*), in contrast to *Th*+ neurons of the SNc (*Figure 3D and F*). Thus, dopamine release from TH+ Pkj cells is unlikely. Exclusion of the first two hypothesized dopamine sources left the nearby monoaminergic LC as a candidate. To verify that the LC projects specifically to cerebellar regions enriched in Drd1+ UBCs, we used a Th-Flpo mouse line. We injected a retrograde adeno-associated virus (AAV2/retro-CAG-fDIO-Cre-EGFP) into lobules IX/X and found green fluorescent protein+ (GFP) LC neurons (*Figure 3G*). Consistent with TH expression in lobule IX/X (*Figure 3D-E* and *Figure 3—figure supplement 1A*), local Pkj cells were also labeled by the injection. The SNc and VTA did not express GFP, further confirming the absence of a TH+ projection from VTA/SNc. Given these anatomical data, the most likely source of dopamine to the Drd1+ UBCs is the LC.

## Tracing inputs of Drd1+ UBCs using trans-synaptic pseudo-typed rabies virus

To assay dopaminergic pre-synaptic inputs to UBCs, we performed trans-synaptic tracing with a modified rabies virus CVS-N2cΔG (*Reardon et al., 2016*) which allowed us to label all pre-synaptic partners of neurons expressing Cre recombinase (*Figure 4A*). The modified rabies virus construct cannot enter cells that do not express the tumor virus receptor A (TVA); it also lacks the glycoprotein necessary to replicate into mature virus that can travel pre-synaptically. We expressed both TVA and the glycoprotein in UBCs using AAVs (AAV1-CAG-Flex-H2B-eGFP-N2cG and AAV1-EF1α-FLEX-GT) in a Drd1-Cre transgenic mouse line. After 4–6 weeks of expression, we injected the modified rabies virus into the same region of the cerebellum. The rabies virus travels monosynaptically to the pre-synaptic partners of local UBCs. Histology was carried out 7–8 days after rabies virus expression. The rabies virus includes a tdT transgene, while the AAVs contain a GFP tag. UBCs initially infected with rabies virus express both GFP and tdT, while cells pre-synaptic to the starter cells express only tdT. To analyze

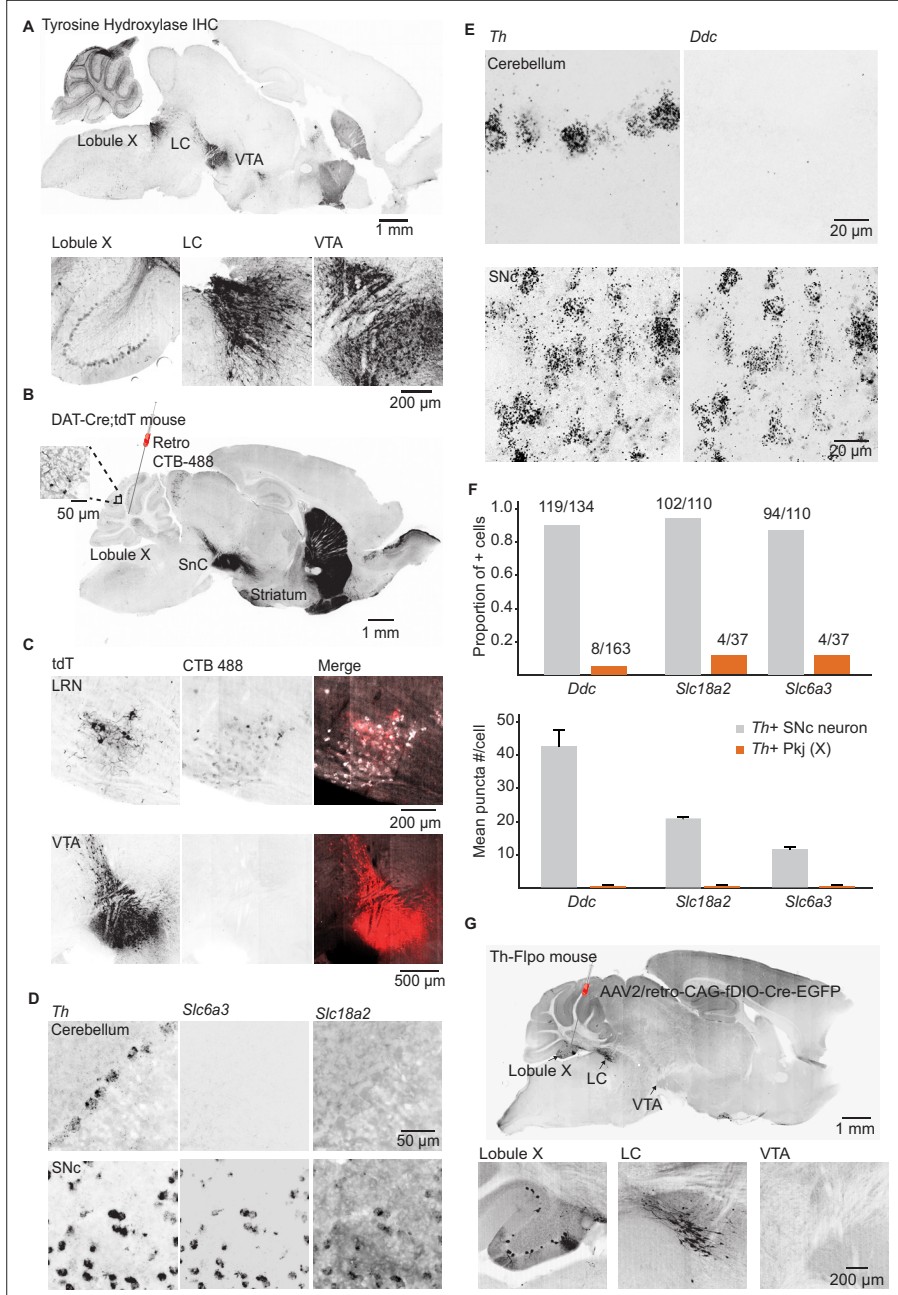

**Figure 3.** Interrogating potential dopamine sources for cerebellar lobules IX/X. (**A**) Top: Parasagittal slice of a mouse brain immunolabeled for tyrosine hydroxylase (TH). Three hypothesized sources of dopamine include Purkinje (Pkj) cells in lobule X of the vermis, locus coeruleus (LC) neurons, and ventral tegmental area (VTA) dopamine neurons. Scale bar, 1 mm. Bottom: Close-up of regions of interest. Scale bar, 200 µm. (**B**) Example retrograde labeling experiment with CTB-488 injected into the cerebellum of a Dat-Cre;tdTomato (tdT) reporter mouse (n = 3 mice). Scale bar, 1 mm. Inset: tdTomato expressing cerebellar fibers are morphologically consistent with mossy fiber terminals and are restricted to the granule layer. Scale bar, 50 µm. (**C**) Close-up images from a retrograde CTB-488 injection. No labeling is observed in the tdT+ neurons in the VTA, in contrast to tdT+ neurons in the lateral reticular nucleus (LRN), a glutamatergic pre-cerebellar brainstem nucleus. Scale bars, 200 µm and 500 µm. (**D**) Top: Confocal images of fluorescent in situ hybridization labeling for tyrosine hydroxylase (*Th*) and dopa decarboxylase (*Ddc*) transcripts in Pkj cells in lobule X. Bottom, images of substantia nigra pars compacta (SNc) neurons co-labeled for Th and Ddc transcripts (n = 2 mice). Scale bar, 20 µm. (**E**) Comparison of TH+ Pkj cells and SNc neurons co-labeled for dopamine active transporter (*Slc6a3*) and vesicular monoamine transporter type 2 (*Slc18a2*) (n = 2 mice). Scale bar, 50 µm. (**F**) Quantification of transcripts involved in synthesis and release of

*Figure 3 continued on next page*

*Figure 3 continued*

dopamine. Top: Few *Th+* Pkj cells express transcripts for *Ddc, Slc18a2,* or *Slc6a3*. Bottom: The few *Th+* Pkj cells that express *Ddc, Slc18a2,* and *Slc6a3* express low transcript numbers in comparison to SNc neurons (n = 2 mice). Gray bars represent counts from SNc dopamine neurons, red bars represent *Th+* Pkj cells. (**G**) Top: Parasagittal section from a Th-Flpo mouse injected with retrograde Flp-dependent virus expressing green fluorescent protein (GFP) (AAV2/retro-CAG-fDIO-Cre-EGFP). Scale bar, 1 mm. Bottom: Close-up of regions of interest. No GFP labeling of VTA/SNc neurons was observed, in contrast to LC neurons and local *Th+* Pkj cells that were retrogradely labeled by GFP (n = 2 mice). Scale bar, 200 µm.

The online version of this article includes the following source data and figure supplement(s) for figure 3:

**Source data 1.** Numerical data for graphs in *Figure 3*.

**Figure supplement 1.** Potential dopamine sources for vestibulo-cerebellum.

**Figure supplement 1—source data 1.** Numerical data for graphs in (*Figure 3—figure supplement 1*).

these data, we used ImageJ-based scripts (*Schindelin et al., 2012*) to quantify the number of UBC starter cells and their pre-synaptic partners after immunohistochemistry to enhance GFP and tdT fluorescence. We did not dissect semicircular canals or otoliths, so they are not included in this analysis, although previous research shows that semicircular canal neurons project directly to mGluR1+ UBCs

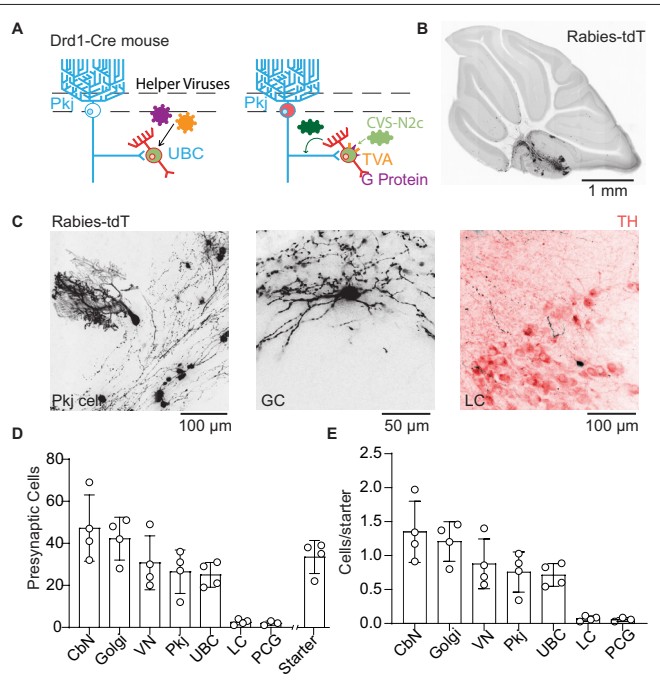

**Figure 4.** Trans-synaptic tracing of inputs to dopamine type 1 receptor expressing unipolar brush cells. (**A**) Schematic of viral targeting for trans-synaptic modified rabies virus tracing. Helper adeno-associated viruses (AAVs) expressing two Cre-dependent constructs, tumor virus receptor A (TVA), and glycoprotein. After 4–6 weeks of helper virus expression in unipolar brush cells (UBCs), we injected modified rabies (CVS-N2cΔG). The rabies virus enters UBCs expressing the TVA receptor. Once expressed in a UBC along with the glycoprotein, the virus can create a functional capsid and travel pre-synaptically. (**B**) Representative image of a sagittal cerebellar slice. Cells infected with the rabies virus express tdTomato (tdT). Scale bar, 1 mm. (**C**) Example cells pre-synaptic to Drd1+ UBCs expressing tdT. Left: Purkinje (Pkj) cell, Center: Golgi cell (GC). Right: locus coeruleus (LC) neuron expressing tdT from the rabies virus, with TH antibody labeling. Scale bars, 50 and 100 µm. (**D**) Number of cells labeled by CVS-N2cΔG rabies virus (n = 4 mice). In addition to UBCs, Pkj cells, and Golgi cells, we found tdT-labeled cells in the cerebellar nuclei (CbN), vestibular nuclei (VN), pontine central gray (PCG), and LC. Starter cells indicate the number of UBCs co-expressing green fluorescent protein (GFP) and tdTomato (tdT). (**E**) Quantification of labeled pre-synaptic cells normalized to the number of starter UBCs.

The online version of this article includes the following source data for figure 4:

**Source data 1.** Numerical data for graphs in *Figure 4*.

(*Balmer and Trussell, 2019*). Our data (n = 4 mice, starter cells 33.5 ± 3.9 /mouse) show pre-synaptic labeling of Pkj cells, Golgi cells, cerebellar nuclei neurons, vestibular nuclei neurons, and a sparse population of LC neurons (*Figure 4B–E*). We did not find any labeling in the midbrain dopaminergic nuclei, even though input tracing using pseudo-typed rabies viral vectors has been used to label monoaminergic inputs (*Schwarz et al., 2015*; *Wall et al., 2013*). It is unknown whether the rabies virus labels certain pre-synaptic inputs preferentially based on properties such as firing rates, axonal length and structure, or neurotransmitter identity (*Rogers and Beier, 2021*). The majority of observed inputs was from nearby cerebellar nuclei cells, consistent with prior reports on their projections back to cerebellar cortex (*Houck and Person, 2015*; *Judd et al., 2021*). Despite low numbers of starter cells, we found sparse labeling of neurons in the LC. These findings are consistent with our anatomical tracing using retro-AAV (*Figure 3G*), confirming the LC as the likely source of dopamine for Drd1+ UBCs.

## Optical measurements of monoamine release from LC fibers onto UBCs

To obtain evidence for functional dopamine release in the cerebellum, we turned to genetically encoded sensors that can directly measure dopamine dynamics with more sensitivity than fast scan cyclic voltammetry and higher temporal resolution than microdialysis (*Patriarchi et al., 2018*; *Sabatini and Tian, 2020*). We injected a Cre-dependent version of GRAB$_{DA2h}$ AAV (*Sun et al., 2018*; *Sun et al., 2020*) into lobule IX/X of the cerebellum of Drd1-Cre animals. We then paired electrical stimulation of cerebellar cortical inputs with two-photon imaging of GRAB$_{DA2h}$ fluorescence in UBCs in the acute slice (*Figure 5A–B*). We found that electrical stimulation of inputs to lobule X (50 stimuli, 50 Hz) elicited an increase in GRAB$_{DA2h}$ fluorescence in UBCs (*Figure 5C*). This increase could be blocked by bath application of Drd2 antagonist, as expected for this sensor (*Figure 5D*) (1 μM L-741,626, n = 10 cells, 4.88-fold reduction in peak amplitude, paired t-test, p=0.002).

While we detected the sensor response with electrical simulation supporting the possibility of dopamine release from axons entering lobule X, this protocol lacks specificity, activating all inputs into lobule X. To selectively activate TH+ inputs from the LC, we crossed Drd1-Cre and Th-Flpo mouse lines to express GRAB$_{DA2h}$ in UBCs and ChR2 in TH+ inputs to the nodulus (*Figure 5E*). We confirmed that our injection of AAV1-Flex-FRT-ChR2-mCherry into the LC labeled axons across the entire cerebellum, including the regions which contain Drd1+ UBCs (*Figure 5F* and *Figure 5—figure supplement 1A*). We imaged GRAB$_{DA2h}$ fluorescence in UBCs in the acute slice while optogenetically activating LC axons (200 pulses, 39 Hz, 2 ms pulses, 460 nm). In each experiment, the absence of TH+ Pkj cell labeling was confirmed to ensure selective LC targeting without overflow of ChR2 expression into the cerebellum. We found time-locked increases of GRAB$_{DA2h}$ fluorescence in UBCs in response to optical stimulation in a subset of UBCs (n = 2 mice, 5/20 cells) (*Figure 5G–H*). Optically evoked GRAB$_{DA2h}$ transients had an average success rate of ~40% across responding cells. Responses to stimulation were bimodal, exhibiting either a substantial change in fluorescence (>0.1 F/F), or no detectable change (*Figure 5G–H*). As a control, we interleaved trials without optogenetic stimulation. In a subset of experiments, we validated the maximal response GRAB$_{DA2h}$ by bath applying Drd2 agonist (1 μM quinpirole) (*Figure 5—figure supplement 1B*). Altogether, these electrical and optical imaging experiments suggest that LC axons release monoamines onto UBCs in lobules IX/X of the cerebellar vermis.

## Electrophysiological recordings of dopamine release from LC fibers

GRAB$_{DA2h}$ is preferentially activated by dopamine, but it maintains a slight sensitivity to NE (dopamine EC$_{50}$ = 0.13 μM, NE EC$_{50}$ = 1.7 μM) (*Sun et al., 2020*). To evaluate whether release of dopamine from LC fibers can activate Drd1 receptors on UBCs, we performed acute slice electrophysiology. Since Drd1 receptors activate Gα$_s$-coupled protein cascades (*Stoof and Kebabian, 1984*; *Surmeier and Kitai, 1993*), and whole-cell recordings can dialyze cytosolic proteins involved in G protein-coupled receptor (GPCR) signaling (*Lahiri and Bevan, 2020*), we decided to use cell-attached recordings to measure potential effect of dopamine release from LC axons. First, we performed cell-attached recordings of Drd1+ UBCs, collected data during a baseline period, and then applied 10 μm SKF81297 (*Figure 5I*). Consistent with our whole-cell current clamp recordings, we measured an increase in spontaneous firing rate in tdT+ UBCs in response to Drd1 agonist (*Figure 5J*) (n = 10 cells, paired Wilcoxon signed rank test, p=0.002). These flow-in data act as a benchmark for the next series of experiments where we optogenetically stimulate LC fibers using a Drd1-Cre; Th-Flpo transgenic cross (*Figure 5K*). We aimed for maximal pharmacological isolation of dopaminergic signaling in these experiments, using

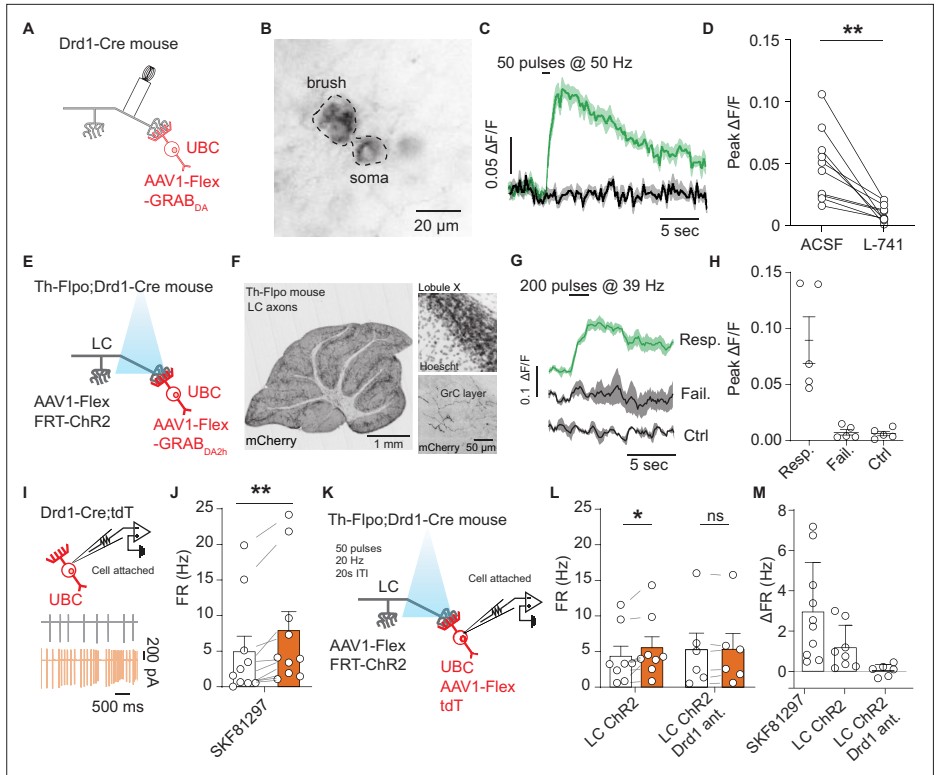

**Figure 5.** Locus coeruleus fibers release dopamine in the cerebellum activating unipolar brush cell dopamine type 1 receptors. (**A**) Schematic of experimental setup, Cre-dependent GRAB_DA2h expressed in unipolar brush cells (UBCs), with a monopolar electrode placed in the white matter of lobule X. (**B**) Two-photon image of UBCs expressing Cre-dependent GRAB_DA2h in a Drd1-Cre cerebellum. (**C**) Sample trace from one cell (five trial average) in response to electrical stimulation before and after application of Drd2 antagonist (1 μM L-741,626). Shaded regions, SEM. (**D**) Group data of peak amplitude of ΔF/F in response to 50 Hz stimulation of axonal cerebellar input before and after flow in of Drd2 antagonist (1 μM L-741,626) (n = 10 cells, paired t-test, p=0.002, **p<0.01). (**E**) Experimental design using dual virus injection to express ChR2 (Flpo-dependent) in locus coeruleus (LC) axons and GRAB_DA2h (Cre-dependent) in UBCs. Light pulses were delivered using 460 nm LED (2 ms pulse width, 200 pulses, 39 Hz). (**F**) Anterograde labeling of axons from the LC to the cerebellum. Left: Injection of AAV1-FLEX-FRT-ChR2-mCherry into the LC shows axons spreading broadly across all layers of the cerebellum (n = 2 mice). Scale bar, 1 mm. Right: Confocal images show LC axons in molecular and granular layer of lobule X. Scale bar, 50 μm. (**G**) Sample traces of changes in GRAB_DA2h fluorescence in response to optical activation of ChR2+ LC axons from a single cell. The three traces correspond to trials with an evoked response, trials with no response, and control trials where no light stimulation was given. Lines represent the mean, shaded regions are ± SEM. (**H**) Summary data from several experiments, 5/20 cells responded to optogenetic activation (six acute slices, n = 2 mice). Error bars represent SEM. (**I**) Diagram of experimental setup, with tdTomato+ (tdT) UBCs recorded in a cell-attached recording configuration. Inset: Traces from an example cell show action potentials before (black) and after (orange) bath application of 10 μM SKF81297. (**J**) Summary data from cell-attached recordings before (white) and after (orange) the application of a Drd1 agonist (n = 10 cells, paired Wilcoxon signed rank test, p=0.002, **p<0.01). (**K**) Schematic of cell-attached recordings paired with selective optogenetic activation of LC fibers. After a baseline period, light stimuli were delivered every 20 s (460 nm, 50 pulses, 20 Hz, 1 ms pulse width). (**L**) Mean firing rates (FRs) before (white) and during (orange) optogenetic stimulation period. Right, same but with Drd1 antagonist in the bath. (**M**) Comparison of changes in FR in response to SKF81297 bath application, optogenetic activation of LC fibers, and optogenetic LC activation in the presence of Drd1 antagonist.

The online version of this article includes the following source data and figure supplement(s) for figure 5:

**Source data 1.** Numerical data for graphs in *Figure 5*.

**Figure supplement 1.** Locus coeruleus fiber location in the cerebellum and validation of GRAB_DA2h.

**Figure supplement 1—source data 1.** Numerical data for graphs in *Figure 5—figure supplement 1*.

a cocktail of receptor antagonists. To avoid activating alpha-adrenergic or beta-adrenergic receptors with LC stimulation (although their expression in UBCs specifically has not been examined), we applied alpha (10 µM prazosin) and beta (10 µM propranolol) adrenergic receptor antagonists. Additionally, since LC neurons have been shown to release glutamate (*Fung et al., 1994*; *Yang et al., 2021*), we blocked AMPA and NMDA receptor mediated excitatory neurotransmission using NBQX (5 µM) and CPP (10 µM). Notably, the blockade of NMDA receptors would be expected to abolish a fraction of Drd1 activation effects (based on our results in *Figure 2H–O*), providing a strong test of the existence of additional mechanisms altering cellular excitability in UBCs downstream of Drd1 stimulation. GPCR cascades have many downstream targets, and so it is not surprising that several mechanisms could be responsible for changes in firing rate. TRPC3 channels were blocked using Pyr-3 (1 µM) to avoid gluta-mate opening TRPC3 channels through mGluR1 activation (*Sekerková et al., 2013*). Pkj cells express adrenergic and Drd2 receptors (*Basile and Dunwiddie, 1984*; *Cutando et al., 2019*; *Kim et al., 2009*; *Lippiello et al., 2015*); to avoid polysynaptic effects from the release of monoamines from LC fibers, we blocked fast inhibitory transmission using gabazine (10 µM) and strychnine (1 µM). For these experiments, to minimize depletion of neurotransmitter from the LC axons, stimulation was delivered at 20 Hz with a 60 s inter-stimulus interval. We measured a small but significant increase in the firing rate of Drd1+ UBCs in response to LC fiber stimulation (*Figure 5L–M*) (1.2 Hz increase, 26% above baseline, n = 8 cells, paired Wilcoxon signed rank test, p=0.02). In a separate set of experiments with Drd1 antagonist (5 µM SCH39166) present in the bath, we measured no statistically significant changes in firing rate.

## Activation of UBCs can pause Pkj cell firing

In order to evaluate the function of Drd1+ UBCs within lobule X of the cerebellar vermis, we trans-fected Drd1-Cre mice with an AAV expressing ChR2-EYFP (yellow fluorescent protein) (AAV5-EF1a-DIO-ChR2-eYFP) and performed cell-attached recordings from Pkj cells (*Figure 6A–B*). We stimulated UBCs with a train of LED pulses (460 nm, 10 Hz, 50 pulses, 1 ms pulse width). Each pulse caused a small burst of action potentials (2–5 APs). UBCs were more likely to respond with multiple spikes to a pulse at the beginning of the train than at the end (*Figure 6B–C*). Pkj cells did not significantly alter their average firing rate on trials with optogenetic stimulation (*Figure 6D*). However, in a subset of Pkj cells, stimulation caused a brief pause in firing during the 500 ms period after the beginning of the optogenetic stimulus (*Figure 6E*). Firing rate was normalized using a z-score. A threshold for changes in firing rate was set at 0.5 z-score SD during the 500 ms period after initial pulse. Cells were classified into 'Bursters', 'Pausers', or 'No response'. The majority of Pkj cells (18/28) did not significantly alter their firing rate, ~ 28% of cells were 'Pausers' (8/28) and 7% of Pkj cells qualified as 'Bursters' (2/28) (*Figure 6F–G*). In a separate group of cells, we performed tight seal recordings from Pkj cells and measured their responses to optogenetic activation of UBCs. Immediately afterward, we broke into the cell to record excitatory and inhibitory post-synaptic currents (EPSCs, –70 mV) (IPSCs,0 mV) in whole-cell voltage clamp mode (*Figure 6H*). We calculated the EPSC/IPSCs charge ratio and plotted it against change in firing rate z-score (*Figure 6I*). Neurons with lower EPSC/IPSC ratios were more likely to pause after the onset of stimulation, suggesting that in these cells the relatively larger inhibi-tory current favored the pause. These data demonstrate that simultaneous activation of Drd1+ UBCs induces a decrease in Pkj cell firing.

## Lobule X Pkj cells directly inhibit UBCs

Pkj cells are canonically known to synapse onto neurons of the cerebellar nuclei (*Telgkamp et al., 2004*), and recent work showed that in the vestibulo-cerebellum Pkj cells make inhibitory connections with nearby granular cells (*Guo et al., 2016*). This local connectivity motif has implications for how Pkj feedback can alter the activity of their own inputs. Given the high density of UBCs in these same regions and the labeling of Pkj cells in our pre-synaptic rabies virus tracing (*Figure 4B–E*), we evalu-ated the possibility of direct synaptic connectivity between Pkj cells and UBCs. We used the Drd1-Cre mouse-line to label UBCs with a Cre-dependent GFP reporter virus and selectively activated Pkj cells using a viral vector expressing ChR2-mCherry driven by the Pcp2 promoter (*El-Shamayleh et al., 2017*; *Figure 7A*). We used epifluorescence to target and patch UBCs with a high chloride ion concentration internal solution. This allowed us to measure IPSCs (*Figure 7B*, *Figure 7—figure supplement 1*) while holding cells at –80 mV. GABA$_A$ currents were isolated using pharmacological blockers of AMPA,

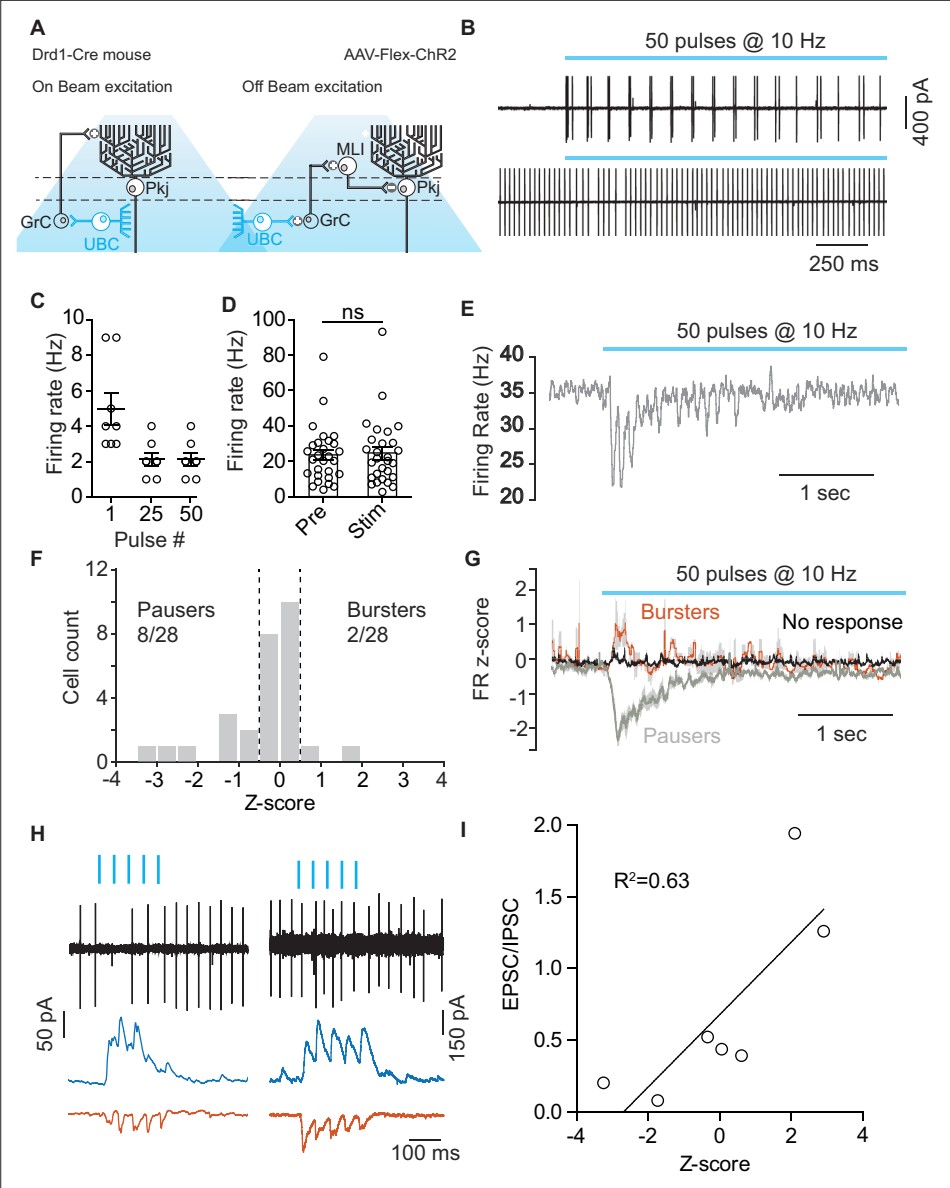

**Figure 6.** Optogenetic activation of unipolar brush cells can modulate Purkinje cell activity. (**A**) Diagram of experimental setup highlighting the two pathways (On-beam, Off-beam) for unipolar brush cells (UBCs) to modulate Purkinje (Pkj) cell firing. Cre-dependent AAV5-EF1a-DIO-ChR2-eYFP was expressed in UBCs using a dopamine type 1 receptor (Drd1)-Cre mouse line. Pkj cells were recorded in cell-attached and/or whole-cell voltage clamp configuration. (**B**) Top: Example trace from a cell-attached recording of a ChR2+ UBC which increased its firing rate (FR) during light stimulation (460 nm, 10 Hz, 50 pulses, 1 ms pulse width). Bottom: Cell-attached recording from a Pkj cell. Recordings were done in the same slice, but not paired. (**C**) The number of action potentials evoked in ChR2+ UBCs by a train of light pulses in response to the 1st, 25th, and 50th individual pulse (n = 8 cells). (**D**) A comparison of average FR of Pkj cells for trials before optogenetic stimulation of UBCs (Pre) and for trials with optogenetic activation (Stim) shows no significant difference. (**E**) The average FR (five trials) of a single Pkj cell during the activation of nearby UBCs. Example cell transiently decreased its FR in response to UBC activation. (**F**) Histogram of average z-scores of FRs during the 500 ms period following the first optogenetic pulse. Dashed lines indicate cut-off for categorization of response as either 'Burster' z-score >0.5 or 'Pauser' z-score <–0.5. (**G**) Average z-score of FRs across cells. 'Pausers' (n = 8 cells) indicates data for cells that had z-score <–0.5, 'Bursters' (n = 2 cells) indicates data for cells that had z-score >0.5, 'No response' (n = 20 cells) include all other cells. Shaded regions represent SEM. (**H**) Cell-attached and post-synaptic current recordings from example Pkj cells; left 'Pauser', right 'Burster'. Top traces show cell-attached recordings before break-in. After break-in inhibitory post-synaptic currents (IPSCs) (blue trace) and excitatory post-synaptic currents (EPSCs) (red

*Figure 6 continued on next page*

*Figure 6 continued*

trace) were recorded. (**I**) FR z-score after optogenetic activation of UBCs plotted against EPSC/IPSC total charge ratio. Data show a correlation between relative inhibitory input and pausing in Pkj cells (linear regression, $R^2$ = 0.63, $F_{1,5}$ = 8.82, p=0.0312).

The online version of this article includes the following source data for figure 6:

**Source data 1.** Numerical data for graphs in *Figure 6*.

NMDA, glycine, and GABA$_B$ receptors (5 µM NBQX, 2 µM CPP, 1 µM strychnine, 1 µM CGP54626). In order to distinguish between direct inhibition from Pkj cells and indirect inhibition from nearby Golgi cells, an mGluR2 agonist was included in the bath to silence Golgi cells (2 µM LY354740) (*Guo et al., 2016*; *Watanabe and Nakanishi, 2003*). To verify that the LED stimulation protocol effectively drove Pkj cell activity, we performed cell-attached recordings. We could reliably increase Pkj cell firing rate during light stimulation (n = 8 cells, 71 Hz ±6.7) (*Figure 7C*). In UBC voltage clamp experiments, we found that 11/24 UBCs received short latency synaptic input from Pkj cells (*Figure 7D*). Cells with successful evoked IPSCs had a per trial success ratio of 0.64 on average (*Figure 7D*). There was

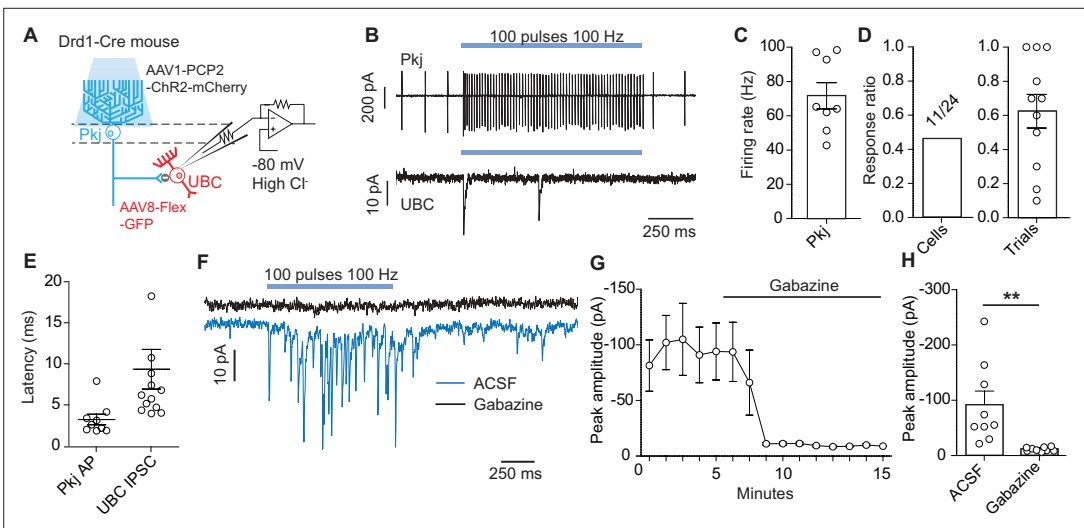

**Figure 7.** Optogenetic activation of Purkinje cells evokes short-latency inhibitory post-synaptic currents in unipolar brush cells. (**A**) Diagram of experimental setup. Using a dopamine type 1 receptor (Drd1)-Cre mouse line a Cre-dependent green fuorescent protein (GFP) was expressed in unipolar brush cells (UBCs), Purkinje (Pkj) cells were transfected with AAV1-Pcp2-ChR2-mCherry. UBCs recorded in whole-cell voltage clamp configuration. Experiments were done with 5 µM NBQX, 2 µM CPP, 1 µM Strychnine, 1 µM CGP54626, and 2 µM LY354740 in the bath. (**B**) Top: Example trace from a cell-attached recording of a mCherry+ Pkj cell which increased its firing rate during light stimulation (460 nm, 100 pulses, 100 Hz, 0.5 ms pulse width). Bottom: Voltage clamp recording of UBC inhibitory post-synaptic currents (IPSCs). Recordings were done in the same slice but not paired. Viral incubation, 3 weeks. (**C**) Average firing rate of Pkj cells during light stimulation (n = 8 cells). (**D**) Ratio of successful IPSC responses. Left (cells): The proportion of cells with IPSCs evoked by optogenetic stimulation of Pkj cells (11/24, n = 24 cells). Right (trials): Ratio of trials in which IPSCs were successfully evoked. Each data point represents the mean from each cell (n = 11 cells). (**E**) The average latency to the first action potential in a Pkj cell (n = 9 cells) or first evoked IPSC in a UBC (n = 11 cells). Latency is calculated from the time of first pulse of 460 nm LED light. Each data point represents the mean from a single cell (5–10 trials). (**F**) Example data from one UBC shows a near complete inhibition of IPSCs (blue trace) after the application of 10 µM gabazine (black trace). Viral incubation, 5 weeks. (**G**) Time course of reduction of IPSC amplitude by gabazine (n = 9 cells). (**H**) Population data comparing peak IPSC amplitude before and after gabazine application (n = 9 cells, paired t-test, p=0.001).

The online version of this article includes the following source data and figure supplement(s) for figure 7:

**Source data 1.** Numerical data for graphs in *Figure 7*.

**Figure supplement 1.** Inhibitory post-synaptic currents in unipolar brush cells in response to Purkinje cell optogenetic activation.

**Figure supplement 1—source data 1.** Numerical data for graphs in *Figure 7—figure supplement 1*.

significant variability in the amplitude (91 pA, ± 24) (*Figure 7H*) and timing (9.4 ms ±2.4) (*Figure 7E*) of optically evoked IPSCs, with some cells exhibiting large currents over the entire stimulation period and others responding with a single large IPSC at the onset of stimulation. However, many UBCs exhibited short latency IPSCs in response to stimulation of Pkj cells (*Figure 7E–F*), on the background of minimal spontaneous IPSCs (*Figure 7—figure supplement 1A*). In a subset of UBCs (n = 9 cells) (*Figure 7F–G*), we flowed in 10 µM gabazine to ensure the measured currents were due to GABA$_A$ channel opening. Application of gabazine abolished all evoked currents (*Figure 7H*).

## Discussion

In this study, we characterized the expression and function of Drd1 receptors in mGluR1+ (ON type) UBCs of the cerebellar vermis. We leveraged the ability to genetically target these UBCs for electrophysiological recordings using the Drd1-Cre mouse line. We found changes in the spontaneous activity of UBCs and their NMDAR-mediated currents by combining electrophysiological recordings with two-photon glutamate uncaging and two-photon imaging. These effects are consistent with the literature on Drd1 receptor function across multiple brain regions (*Chen et al., 2004*; *Chen et al., 2007*; *Lahiri and Bevan, 2020*). We evaluated several distinct hypotheses for the source of dopamine to the cerebellum and found that neurons in the LC project to cerebellar regions enriched in Drd1+ UBCs. Optogenetic activation of LC fibers elicited reliable responses in the dopaminergic optical sensor GRAB$_{DA2h}$ which is consistent with local dopamine co-release from TH+ LC inputs. We found that UBCs receive direct inhibition from local Pkj cells and that the activation of UBCs can alter the firing rate of Pkj cells, forming a closed reciprocal loop. These findings expand our knowledge about neuromodulation in the cerebellum and reveal a new function for LC inputs in cerebellar processing. The work presented here focused on discovering and validating the functional existence of a connection between the LC and Drd1+ UBCs. We relied on slice physiology and viral techniques that favored specificity of labeling. However, this means that the activity rates, connectivity ratios, and amplitude of effects are likely different in the intact circuit in vivo. Specifically, inferences from acute slice experiments are likely to underestimate connectivity and effect magnitudes, with the benefit of increased specificity. Future experiments will need to use a variety of in vivo recording or imaging techniques to revisit the functions of LC-UBC connectivity in intact circuits.

While the cerebellum is known to receive dense innervation from several neuromodulatory nuclei (*Li et al., 2014*; *Nelson et al., 1997*; *Schweighofer et al., 2004*), questions about the role of these neuromodulators have been difficult to address. The function of dopamine in the cerebellum has been particularly elusive due to disagreement about the location of receptors and potential sources of dopamine (*Hurley et al., 2003*; *Kim et al., 2009*; *Locke et al., 2018*; *Mansour et al., 1991*). To evaluate cerebellar dopamine receptor expression, we used a combination of transgenic tools and FISH, characterizing the distribution of *Drd1a* mRNA in mGluR1a+ UBCs from the Drd1-Cre line crossed to a tdT reporter. Most Drd1 UBCs express a low amount of *Drd1* transcripts. Since UBCs are compact cells with small somato-dendritic compartments, a low transcript number of *Drd1a* mRNA could produce sufficient receptor levels to impact UBC activity. We evaluated the consequences of Drd1 activation by recording the activity of Drd1+ UBCs before and after activating Drd1 receptors. Most tdT+ UBCs (70%) increased their firing rate in response to Drd1 activation, as has been reported for Drd1-mediated effects on striatal spiny projection neurons (*Lahiri and Bevan, 2020*). Quiescent neurons characterized by hyperpolarized resting membrane potential did not initiate activity in response to pharmacological stimulus. This observation suggests several non-exclusive possibilities. First, Drd1 receptor–mediated depolarization may be insignificant in some UBCs. Second, the effect of Drd1 agonism is, at least in part, dependent on ionic currents that are not active below a certain membrane potential threshold. Third, the effect of Drd1 activation could be unable to overcome the ionic currents that regulate the 'down' state in UBCs. One intriguing possibility is that previously described bimodal states in UBC membrane potential (*Diana et al., 2007*) map onto distinct responses to neuromodulatory inputs.

Based on previous literature on the mechanisms of dopamine neuromodulation in the prefrontal cortex (*Chen et al., 2004*), we evaluated NMDAR-mediated currents in response to mossy fiber electrical stimulation, finding an increase in NMDAR currents in the presence of a Drd1 agonist. We relied on two-photon glutamate uncaging and imaging combined with electrophysiology to confirm the post-synaptic mechanism for Drd1-dependent increase in NMDAR currents in UBCs. As

compact neurons, UBCs have relatively small total NMDAR currents (*Billups et al., 2002*; *Rossi et al., 1995*), yet this increase could have wide ranging consequences for cellular activity. The enhancement in protein kinase A activity, a consequence of Drd1 receptor activation, can promote calcium flux through NMDARs (*Skeberdis et al., 2006*). Recent work has shown that the brush of UBCs is exposed to tonic levels of glutamate released from mossy fiber terminals in a spike-independent manner (*Balmer et al., 2021*). This basal glutamate concentration in the synaptic cleft serves a role in calibrating synaptic responses of both ON and OFF UBCs and can be disrupted by blocking glutamate re-uptake pharmacologically. Given that UBCs have spontaneous firing rate, this depolarization paired with basal glutamate suggests potential activation of NMDA receptors even in the absence of synaptic input. Drd1 effects, mediated by $G\alpha_s$-coupled pathways, have diverse impacts on neuronal channels in other cell types, raising the likelihood that currents in UBCs are also altered by Drd1 activation. Of special note for Drd1+ UBCs, dopamine might enhance mGluR1-mediated transient receptor potential channel (TRPC3) currents (*Sekerková et al., 2013*), as has been previously demonstrated for striatal cholinergic interneurons (*Chuhma et al., 2018*). If so, this mechanism would have synergistic effects with NMDAR-mediated current enhancement in response to Drd1 receptor activation. We next investigated three potential non-exclusive sources of dopamine to the cerebellum: the SNc/VTA, local TH+ Pkj cells, and the LC. Consistent with some (*Wagner et al., 2017*) but not all previous reports (*Melchitzky and Lewis, 2000*; *Panagopoulos et al., 1991*), we found no evidence of projections from the SNc/VTA. While we were able to confirm TH expression in nearby Pkj cells (*Locke et al., 2020*; *Takada et al., 1993*), FISH analyses of *Ddc*, *Slc18a2*, and *Slc6a3* in the *Th*+ neurons demonstrate that these neurons lack the molecular machinery for releasing dopamine. Previous work has found that while Pkj cells expressed TH, this protein is maintained in an unphosphorylated state, unable to produce L-DOPA (*Lee et al., 2006*; *Sawada et al., 2004*). Given the absence of L-DOPA synthesis, the function of TH in Pkj cells currently remains unknown. However, there are other neuronal classes that express TH but lack complementary proteins for dopamine release, including a subpopulation of olfactory bulb neurons (*Chand et al., 2015*), striatal GABAergic interneurons (*Weihe et al., 2006*; *Xenias et al., 2015*), and cortical interneurons (*Asmus et al., 2008*). As the final potential source of dopamine for UBCs, we evaluated the LC, known to project to the cerebellum (*Bloom et al., 1971*; *Schwarz et al., 2015*) and to co-release NE and dopamine in other brain regions. Dopamine released from LC fibers in the hippocampus increases field EPSCs and contributes to memory retention (*Kempadoo et al., 2016*; *Takeuchi et al., 2016*). In the paraventricular nucleus of the thalamus, dopamine released by LC axons activates Drd2 receptors to decrease inhibitory input amplitude (*Beas et al., 2018*). To test whether LC projections release dopamine onto UBCs, we used two-photon imaging of genetically encoded sensor GRAB$_{DA2h}$. Electrical or optogenetic activation of cerebellar LC axons increased GRAB$_{DA2h}$ fluorescence in UBCs. Additional studies are needed to examine whether UBCs also express multiple GPCRs that can be concurrently activated by LC inputs. While LC neurons send projections across the cerebellar cortex, they are comparatively less dense within lobule X. This is consistent between our data and another recent report (*Carlson et al., 2021*). Although this axonal distribution circumscribes the maximum impact of LC neuromodulatory processes in this lobule, confocal images reveal the consistent presence of LC axons in both the granular and molecular layers of lobule X, and GRAB$_{DA2h}$ experiments show that LC axons can directly release monoamines onto lobule X UBCs. While GRAB$_{DA2h}$ is highly specific for dopamine over NE (DA EC$_{50}$ = 0.13 μM, NE EC$_{50}$ = 1.7 μM) (*Sun et al., 2020*), we cannot exclude the possibility of NE release from LC axons using the results from optical sensors. No prior research has focused on adrenergic receptor expression in UBCs, so it is possible that LC axons could mediate their effect on UBCs through NE, in addition to dopamine. To address the ambiguity of potential co-release, we performed cell-attached recordings in UBCs, pharmacologically isolating Drd1 receptors, and found an increase in firing rate after stimulation of LC fibers. To avoid effects of co-release of glutamate from LC axons in these experiments, we had to block NMDA receptors, removing one downstream mechanism of Drd1 activation in UBCs. In agreement with experiments examining Drd1 activation across different brain regions, the persistence of Drd1 receptor–mediated effects suggests that there are likely multiple intracellular mechanisms governing UBC's response to dopamine. One such potential mechanism could involve voltage-gated calcium currents. UBC expresses both high and low voltage-activated calcium channels (*Birnstiel et al., 2009*) known to respond to activation of Drd1 in other brain regions (*Hernández-López et al., 1997*; *Liu et al., 2004*; *Surmeier et al., 1995*).

The Drd1-Cre mouse line is one of the few tools that allow us to selectively label subpopulations of UBCs. Using ChR2, we took advantage of this mouse line to interrogate the potential function of UBCs within vestibulo-cerebellar circuits. Despite the utility of acute slice preparation as a tool in electrophysiology, the approach has important limitations, undercounting axonal inputs and impacting neural activity in polysynaptic neuronal circuits. For this reason, we focused on Pkj cell firing rate changes on a short time scale following optogenetic activation (500 ms), aiming to evaluate proximal effects downstream of activating a small neuronal ensemble. The results from our experiments demonstrate that activation of UBCs can induce pausing in a subset of local Pkj cells. Given our data examining Pkj cell E/I balance during UBC optogenetic stimulation, changes in Pkj activity are likely explained by differential activation of On and Off parallel fiber beams (*Dizon and Khodakhah, 2011*). 'On beam' parallel fibers, originating from the granule cells directly below a Pkj neuron, make excitatory synapses onto that Pkj cell. Meanwhile, granule cells that are not directly below a Pkj cell form 'Off beam' parallel fibers, activating molecular layer interneurons and disynaptically inhibiting adjacent Pkj cells (*Figure 5A*). Both 'On' and 'Off' beam excitation are likely to occur simultaneously. Due to the spatial distribution of UBCs in the granular layer and whole-field optogenetic activation protocol, 'Off beam' parallel fiber excitation is favored in the slice configuration. How likely is it that similar concurrent activation of UBCs occurs in vivo? The majority of UBCs expresses mGluR1 receptors and fire bursts of action potentials in response to glutamate release from mossy fibers (*Zampini et al., 2016*). One possibility is that due to recurrent connections between UBCs, a burst of action potentials in one UBC could cause a chain of activity leading to near simultaneous granular layer activity across a large area.

In addition to functionally evaluating the evidence for Pkj-UBC connectivity to understand the placement of Drd1+ UBCs within vestibulo-cerebellar circuits, our trans-synaptic rabies virus tracing data confirmed the labeling of Pkj cells, while also revealing the neuromodulatory inputs from the LC. Recent work has found that Pkj cells in the vestibulo-cerebellum synapse onto nearby granule cells (*Guo et al., 2016*), but it remained unknown whether local UBCs receive feedback input from Pkj cells. Notably, Pkj cell feedback to granule cells appears to be restricted to the vestibular regions of the cerebellar cortex, with important implications for the recurrent activity of Pkj cells in these lobules. Our findings expand the recurrent vestibulo-cerebellar circuit to include UBCs. Since UBCs diverge in the granular layer, any circuit level effects from feedback inhibition via Pkj cells is likely to be amplified. These inhibitory connections have also been described in parallel work by another lab (*Guo et al., 2021*) for mGluR1+ UBCs. LC neurons are activated in a wide range of behaviors, including deployment of attention (*Aston-Jones et al., 1999*), sleep-wake transitions (*Hayat et al., 2020*), and locomotion (*Xiang et al., 2019*). There is longstanding evidence of the LC modulating Pkj cell activity through the release of NE (*Basile and Dunwiddie, 1984*; *Lippiello et al., 2015*), and recent work shows adrenergic signaling in Golgi cells as well (*Lanore et al., 2019*). Our findings suggest that the LC could release dopamine onto other cells in the cerebellum, such as Drd2 expressing Pkj cells (*Kim et al., 2009*; *Mengod et al., 1989*), in addition to UBCs. Multiple links between LC and vestibular processing have been reported. First, LC neurons respond to rotations of the neck (*Manzoni et al., 2009*; *Pompeiano et al., 1991*). Second, vestibulo-cerebellar Pkj cells project to the LC as well as the vestibular nuclei (*Schwarz et al., 2015*). Finally, vestibular nuclei make synaptic connections in the cerebellum (*Balmer and Trussell, 2019*; *Päällysaho et al., 1991*) and receive projections from the LC (*Schuerger and Balaban, 1993*). This interconnectivity between the three regions suggests that NE and dopamine from LC fibers could play a role in vestibular processing.

UBCs have been found in three distinct brain circuits implicated in the cancelation of self-generated cues due to an animal's movement (*Requarth and Sawtell, 2014*; *Singla et al., 2017*; *Warren and Sawtell, 2016*). Their position in the input layer of the cerebellum means that their activity can significantly alter mossy fiber signals to the cerebellum. In the mouse cerebellum, Drd1 expressing UBCs reside in regions involved in processing vestibular signals and eye movements. Since we find that Drd1 receptors are expressed predominantly in ON UBCs, dopamine release from LC axons could coordinate with vestibular mossy fiber inputs to enhance signals arising directly from the semicircular canals (*Balmer and Trussell, 2019*). Leveraging genetic access to a subpopulation of UBCs, we have characterized a new recurrent subcircuit between Pkj cells and UBCs, which has the potential to significantly regulate the output of the vestibulo-cerebellum. Altogether, this work expands our knowledge of cerebellar circuitry, resolves longstanding questions about the ability of TH+ Pkj cells to release dopamine, and points to the LC as the source of dopamine to the cerebellum.

# Materials and methods

## Mouse Lines

All experiments were performed under the guidelines set by Northwestern University Institutional Animal Care and Use Committee. All mice used were of the C57BL/6 background (P20-P80). Mice of both sexes were used; sex was tracked for every experiment, and data were analyzed for any differences. Several transgenic lines were used for this project. To visualize Drd1-positive cells, we used a heterozygous mouse line with an insertion of Cre recombinase at the Drd1a locus (Drd1a-Cre, GENSAT, founder line EY262) referred to as Drd1-Cre (*Gong et al., 2003*). In order to label cells expressing dopamine transporter (DAT), we used knock-in mice expressing Cre recombinase under DAT promoter, B6.SJL-Slc6a3$^{tm1.1(cre)Bkmn}$/J mice (*DAT$^{i-Cre}$*) (Jackson Lab, No. 006660, Bell Harbor, MA) (*Bäckman et al., 2006*). To label TH-positive neurons, we used a Th-Flpo developed by the lab of Dr. Raj Awatramani (*Poulin et al., 2018*). Depending on experiments, these Cre lines were either injected with a viral vector or crossed with a floxed tdT reporter mouse line (Ai14, Jackson Lab, No. 007914) (*Madisen et al., 2010*).

## Viral vectors

Viruses were acquired from Addgene, UNC vector core, Columbia vector core, Salk Institute, and WZ Biosciences. For stimulation of Drd1-Cre+ UBCs, we injected AAV5-EF1a-DIO-hChR2(H134R)-eYFP-WPRE-hGH ($7.2 \times 10^{12}$ vg/ml, Addgene, #20298). For experiments with optogenetic stimulation of Pkj cells, we injected AAV1-PCP2-CHR2-mCherry ($7.2 \times 10^{12}$ vg/ml) (*El-Shamayleh et al., 2017*) and AAV8-EF1a-Flex-GFP ($6.2 \times 10^{12}$ vg/ml, UNC vector core). For pseudo-typed rabies tracing experiments, the viruses were AAV1-CAG-Flex-H2B-eGFP-N2c(G) ($2.5 \times 10^{12}$ gc/ml, Columbia vector core), AAV1-EF1α-FLEX-GT ($4.3 \times 10^{11}$ gc/ml, Salk Institute viral vector core), and CVS-N2cΔG tdT EnvA (Columbia vector core). For retrograde labeling of TH fibers to the cerebellum, we used AAV2/retro-CAG-fDIO-Cre-EGFP (a gift from the laboratory of Bernardo Sabatini). For anterograde labeling of LC fibers, we injected AAV1-CAG-FLEXFRT-ChR2(H134R)-mCherry (75470, $7 \times 10^{12}$ vg/ml, Addgene). For imaging dopamine dynamics, we used AAV9-hSyn-DIO-DA4.3, (GRAB$_{DA2h}$, 20191018, $2.7 \times 10^{13}$ vg/ml, WZ Biosciences; plasmid, gift from the laboratory of Yulong Li).

## Surgical procedures

Animals were anesthetized using an isoflurane vaporizer, with oxygen flow rate maintained at 1.00 l/min. Induction was done with 2–3% isoflurane and maintenance at 1.5–2.5%. Tail pinches were used to ensure the effectiveness of anesthesia. Animal was mounted on a stereotaxic frame (David Kopf Instruments, Tujunga, CA), ear bars were inserted, and a chemical depilator was used to expose the skin. Skin was cleaned with iodine followed by alcohol, using aseptic technique. Injections of AAV viral vectors or CTB-488 (Thermo Fisher, C34775) were made using a pulled glass pipette and a UltraMicroPump controller (World Precision Instruments, Sarasota, FL) at a rate of 100 nl/min. Rabies virus injections were performed in a BSL2 hood. Cerebellar stereotaxic coordinates: AP: –2.5 mm ML: 0 mm, from lambda, DV: –2.5 mm from surface of brain. Coordinates for the LC: AP: –0.9 mm ML: 0.9 mm, from lambda, DV: –3.2 mm from surface of brain. All animals received an I.P. injection of ketoprofen (0.1 mg/20 g) 0 hr, 24 hr, and 48 hr after surgery, with approved post-surgical monitoring. Mice recovered for 7 days for retrograde CTB-488 injections and 2–6 weeks for viral vector injections. For modified rabies virus tracing experiments, we allowed the helper viruses to express for 4–6 weeks before injecting rabies virus.

## Immunohistochemistry

Animals were anesthetized using isoflurane and then transcardially perfused with 4% paraformaldehyde (PFA). The brain was extracted and post-fixed at 4°C in 4% PFA overnight, transferred to phosphate buffered saline (PBS), and stored at 4°C. The brain was mounted onto a Leica VT1000s vibratome and 50 µm thick sections were made. Sections were stored in well plates in PBS at 4°C. Slices were permeabilized in 0.2% Triton-X, then blocked in 10% BSA with 0.05% Triton-X, and incubated in primary antibody for 24–48 hr at 4°C in 0.2% Triton-X. The tissue underwent three 10-min rinse steps in PBS and was then incubated in a secondary antibody for 2 hr at RT. Primary antibodies: Rabbit anti-RFP (1:1000, Rockland, Cat No. 600-401-379), Chicken anti-GFP (1:1000, Abcam, ab13970), Rabbit anti-Calretinin (1:500, Swant, CR7697), Mouse anti-mGluR1a (1:500, BD

Biosciences, G209-2048), Sheep anti-TH (1:1000, Abcam, ab113), and Chicken anti-Tbr2 (1:1000, Sigma-Aldrich, AB15894). Secondary antibodies: Goat Anti-Mouse 647 (1:1000, Life Technologies, A-21236), Goat Anti-Rabbit 488 (1:1000, Life Technologies A-11034), Donkey-anti-Sheep 647 (1:1000, Thermo Fisher Scientific, A-21448), Goat anti-Chicken IgY (H + L) secondary antibody, and Alexa Fluor 488 (1:1000, Life Technologies A-11039). Following three rinse steps in PBS, samples were mounted onto Superfrost Plus slides (ThermoFisher Scientific, Waltham, MA) air-dried and cover slipped under glycerol:TBS (3:1) with Hoechst 33,342 (1:1000; ThermoFisher Scientific). Slides were imaged at ×10 magnification on a slide scanning microscope Olympus VS120 slide scanning microscope (Olympus, Waltham, MA). For IHC, quantification slides were imaged using a Leica SP5 confocal microscope (Leica Microsystems) with a ×40/63 objective. Z stacks at 2–5 µm step sizes were acquired, and images were analyzed in ImageJ/FIJI (FIJI is just ImageJ) (*Schindelin et al., 2012*).

## Fluorescence in situ hybridization

Following anesthetization using isoflurane, the mouse was decapitated, and the brain was quickly removed, placed into optimal cutting temperature compound (OCT) and frozen on dry ice. Samples were stored in –80°C freezer for at least 12 hr and no longer than 7 days. Tissue was sliced on a cryostat (Leica CM1850) at a thickness of 20 µm and mounted onto Superfrost Plus slides (ThermoFisher Scientific, Waltham, MA). Slices were placed in a slide box and in a Freezer safe Ziploc bag and placed in a –80°C freezer overnight before processing. Samples were fixed with 4% PFA in 0.1 M PBS at 4°C, processed according to RNAscope Fluorescent Multiplex Assay manual for fresh frozen tissue (Advanced Cell Diagnostics, Newark, CA) and coverslipped using ProLong Gold antifade reagent with DAPI (Molecular Probes). Several probes were used including: *tdTomato/tdT* (ACDBio Cat No. 317041), *Drd1a* (ACDBio, Cat No. 406491-C2), *Th* (ACDBio, Cat No. 317621-C2), *Slc18a2* (ACDBio Cat No. 425331-C3), *Ddc* (ACDBio, Cat No. 318681), *Slc6a3* (ACDBio, Cat No. 315441), and *Slc17a7* (ACDBio, Cat No. 416631). Slides were imaged on a Leica SP5 confocal in two channels using a ×63 objective digital zoom ×1.5 with 0.5 micron z steps (512 × 512 pixels). FISH images were analyzed using ImageJ (*Schindelin et al., 2012*). Images were background subtracted and thresholded. We then used 3D objects counter (*Bolte and Cordelières, 2006*) to create a 3D object. The 3D ROI manager (*Ollion et al., 2013*) plugin was used to create ROI. The tdT channel was used to as a label for UBC soma and to establish ROI. Once ROIs were acquired, we quantified the overlap of signals by counting the number of *Drd1a* puncta within each cell.

## Acute slice preparation

Mice (P20-60) were deeply anesthetized with isoflurane, followed by a transcardial perfusion using 34°C (*Huang and Uusisaari, 2013*) ACSF containing (mM) 127 NaCl, 2.5 KCl, 1.25 NaH$_2$PO$_4$, 25 NaHCO$_3$, 20 glucose, 2 CaCl$_2$, and 1 MgCl$_2$. After perfusion mice were decapitated, brain was removed, blocked, mounted, and placed into a chamber containing 34°C ACSF oxygenated with 95% O$_2$ and 5% CO$_2$. Parasagittal 300 µm cerebellar brain slices were made using a vibratome (Leica VT1200s). Slices were transferred to a holding chamber containing 34°C ACSF oxygenated with 95% O$_2$, 5% CO$_2$, and were incubated for 30 min before being cooled to room temperature (22–24°C).

## Electrophysiology

All recordings were performed on an acute slice electrophysiology system (Scientifica, UK); neurons were visualized using a ×60 water-immersion objective (Olympus, Tokyo, Japan) imaged using a QIClick microscope camera (QImaging, Surrey, Canada), using differential interference contrast (DIC). Slices were placed in the recording chamber and perfused with oxygenated ACSF at a rate of 2 ml/min, with temperature as noted. For UBC recordings, cells expressing tdT in a Drd1-Cre; tdT cross were targeted using epifluorescence illumination from a CoolLED pe4000 system (CoolLED Ltd., Andover, UK). Whole-cell patch electrodes were pulled from borosilicate capillary glass to have a resistance of 2.5–5 MΩ for UBC recordings and 1–2 MΩ for Pkj recordings. All recordings were digitized at 10–20 kHz and filtered at 3–4 kHz using a Multiclamp 700b amplifier (Axon Instruments, Union City, CA), acquired using a version of the MATLAB-based (MathWorks, Natick, MA) acquisition suite, ScanImage (*Pologruto et al., 2003*).

## Current clamp recordings

UBCs were targeted using a Drd1-Cre;tdT transgenic mouse line (P20-P40). Slices were perfused with ACSF at RT (22–24°C). For current clamp recordings, any cell with a series resistance >40 MΩ or an action potential peak below 10 mV was excluded from analysis. Intracellular solution used was (in mM) 135 K-gluconate, 4 KCl, 10 HEPES, 10 Na-phosphocreatine, 4 MgATP, 0.4 Na$_2$GTP, 1 EGTA, 20 µM Alexa 488 (pH 7.2–7.3, 295–298 mOsm/L). A measured liquid junction potential of ~12 mV was corrected for before analysis. Every sweep was 15 s long, with a 20 s inter-trial interval (ITI) between each trial. Within each trial, a 500 ms hyperpolarizing step 10–20 pA was injected in order to measure input resistance. In cells that did not have spontaneous firing, we injected a 500 ms long depolarizing pulse to assess cell health and ability to fire action potentials. Both the period of the hyperpolarizing pulse and depolarizing pulse were excluded from analysis of spontaneous firing rate. To calculate resting membrane potential, we truncated action potentials with a –10 mV cut off, applied a median filter over a period of 5 s, and calculated the average over that period. There was no holding current imposed on cells except for the hyperpolarizing pulse used to measure relative input resistance. UBCs rest at different membrane potentials, for this reason the input resistance measured is relative. We analyzed spontaneous activity, passive membrane properties, and action potential shape. After break-in, we waited ~13 min period before acquiring baseline firing properties of the UBCs. The baseline period lasted ~7 min (20 trials) before we applied SKF81297. To activate Drd1 receptors, we puff applied SKF81297 (500 µM) using a 300 ms long PicoSpritzer puff (Parker Hannifin, Hollis, NH). The drug was applied once per trial (20 s ITI) for a 10 min period (30 trials). We continued recording after drug application for a 10-min (30 trials) washout period. The glass puff pipette was pulled to have a ~5 µm opening. The puff pipette was placed 20–40 µm above tissue surface and 50–75 µm away from the patched cell on the x-axis. Care was taken to ensure there was no movement of the slice due to the puff. After the recording, each UBC was imaged to verify that it was filled with Alexa 488 dye, and then a ×10 magnification image was taken to record location within the cerebellum.

## Voltage clamp recordings

UBCs were targeted using a Drd1-Cre;tdT transgenic mouse line (P20-P40). Voltage clamp recordings were performed in ACSF (32–34°C). Cells were excluded if they had a series resistance >25 MΩ or if series resistance changed >20% during the recording. We did not use series resistance compensation. The internal electrode solution contained (in mM): 125 CsMeSO$_3$, 5 HEPES-Cs, 0.4 EGTA, 10 Phosphocreatine disodium salt, 2 ATP-Mg, 0.5 GTP-Mg, 5 TEA-Cl, 2 QX314-Cl, and 20 µM Alexa 488 (pH 7.2–7.3, 295–298 mOsm/L). We corrected for liquid junction potential (~8 mV) before acquiring data. In order to isolate NMDAR currents, 1 µM Gabazine (Tocris, 1262), 1 µM NBQX (Tocris, 0373), 1 µM strychnine (Sigma-Aldrich, S8753) were added to the ACSF bath. Cells were held at +40 mV, and we used 0 mM MgCl added to ACSF to minimize the Mg block of NMDA receptors. A tungsten bipolar electrode (CBAPB75, FHC Inc) was lowered into nearby white matter at least 100 µm away from UBC being recorded. Each electrical stimulation consisted of 5 pulses, 200 Hz, 100–150 µA, 50–80 µs pulse width. Each trial was 7 s long, with an ITI of 30 s. Every trial included a 500 ms long 20 mV hyperpolarizing pulse to measure series resistance. We acquired current traces during a baseline period of 20 min (40 trials), followed by a flow-in of SKF81297 (10 µM) for 15 min (30 trials) before washing out the drug.

## Optogenetic stimulation of Drd1-Cre UBCs

Pkj cells within 100 µm of ChR2-EYFP expressing UBCs were targeted for recording. Recordings were performed in ACSF (32–34°C). We recorded baseline firing rate of Purkinje cells in cell-attached mode, loose seal (50–200 M Ohms). To stimulate ChR2 in UBCs, we used a train of light pulses (1 ms, 10–20 Hz, 20 pulses, 460 nM, 2–5 mW) delivered through a ×60 water-immersion objective (Olympus, Tokyo, Japan) with CoolLED PE4000 widefield illumination (CoolLED Ltd., Andover, UK). Every trial was 15 s long, with an ITI of 20 s. After an ~7 min (20 trials) baseline period, we stimulated ChR2 for 30 trials. Afterwards, a 10-min (30 trials) recording period with no stimulation was acquired. For each slice, we verified that we could reliably evoke action potentials in nearby yellow fluorescent protein+ (YFP) UBCs with this stimulation protocol.

For the experiments with EPSC and IPSC recordings, we used an internal solution consisting of 125 CsMeSO$_3$, 5 HEPES-Cs, 0.4 EGTA, 10 Phosphocreatine disodium salt, 2 ATP-Mg, 0.5 GTP-Mg, 5

TEA-Cl, 2 QX314-Cl, and 20 µM Alexa 488 (pH 7.2–7.3, 295–298 mOsm/L). We corrected for liquid junction potential (~8 mV). We first achieved a Giga-Ohm seal on the Pkj cell before recording cell-attached action potentials, to evaluate the response of Pkj cells to UBC optogenetic activation before breaking into the Pkj cell and recording synaptic currents. We recorded EPSCs while holding the cell at –70 mV (five trials), then slowly increased the holding voltage to 0 mV and recorded IPSCs (five trials).

## IPSC recordings in UBCs

Drd1-Cre animals were injected with AAV1-PCP2-ChR2-mCherry along with AAV8-EF1a-Flex-GFP. Acutely prepared brain slices were perfused with ACSF at RT (22–24°C) to slow down spontaneous firing rate of Pkj cells. We used epifluorescence to verify expression of mCherry in nearby Pkj cells and used very brief GFP illumination to target UBCs for patch clamping without stimulating ChR2. We used an internal solution with a high concentration of Cl$^-$ ions (in mM): 150 CsCl, 4 NaCl, 0.5 CaCl$_2$, 10 HEPES-Cs, 5 EGTA-Cs, 10 Phosphocreatine disodium salt, 2 ATP-Mg, 0.5 GTP-Na, 2 QX314-Br, and 20 µM Alexa 488 (pH 7.2–7.3, 295–298 mOsm/L). We corrected for liquid junction potential (~3 mV). Starting 5 min after break-in, we recorded 10 trials/UBC, where every trial was 15 s long, with an ITI of 60 s. During every trial, we delivered an optogenetic stimulus train (1 ms, 100 Hz, 100 pulses, 460 nM, 2–5 mW) through a ×60 water-immersion objective (Olympus, Tokyo, Japan) with CoolLED illumination PE4000. For a subset of experiments, after five trials we added 10 µM Gabazine to the bath. For every slice, cell-attached recordings were used to verify efficient optogenetic stimulation of Pkj cell firing.

## Cell-attached recordings

UBCs were targeted using a Drd1-Cre;tdT transgenic mouse line (P20-P40). Cell-attached recordings were performed in ACSF (32–34°C). Internal electrode solution contained (in mM): 125 CsMeSO$_3$, 5 HEPES-Cs, 0.4 EGTA, 10 Phosphocreatine disodium salt, 2 ATP-Mg, 0.5 GTP-Mg, 5 TEA-Cl, 2 QX314-Cl, and 20 µM Alexa 488 (pH 7.2–7.3, 295–298 mOsm/L). Every sweep was 15 s long, ITI 20 s. After an ~16-min (50 trials) baseline period, we flowed in 10 µM SKF81297 (Tocris 1447) into the bath. Cells showing spontaneous firing were included in analysis for continual monitoring of cell health. For experiments using LC optogenetic stimulation, we included the following in the bath: Prazosin (10 µM), propranolol (10 µM), NBQX (5 µM), CPP (10 µM), Pyr-3 (1 µM), gabazine (10 µM), and strychnine (1 µM). For recordings in the presence of Drd1 antagonist, 5 µM SCH39166 was present in the bath. During every trial, we delivered an optogenetic stimulus train (1 ms, 39 Hz, 200 pulses, 460 nM, 1–2 mW) through a ×60 water-immersion objective (Olympus, Tokyo, Japan) with CoolLED illumination PE300, with a 20 s ITI.

## Two-photon imaging and glutamate uncaging experiments

Slices were perfused with ACSF at RT (22–24°C). Experiments were done on a modified acute slice electrophysiology Scientifica microscope using a ×60 water immersion objective. Two mode-locked Ti:Sapphire lasers (Mai Tai eHP and Mai Tai eHP DS, Newport) were used for two-photon fluorescence imaging and for uncaging, with the wavelengths of 910 nm and 725 nm, respectively (*Banala et al., 2018*; *Chen et al., 2020*; *Kozorovitskiy et al., 2015*; *Wu et al., 2021*; *Xiao et al., 2018*). The location of the laser beam was controlled by a 2D galvanometer scanning mirror system (HSA Galvo 8315 K, Cambridge Technology). Fluorescence emission was passed through a dichroic beamsplitter (FF670-SDi01−26 × 38, Semrock) and a bandpass filter (FF02-520/28, Semrock). Emission light was collected by two photomultiplier tubes (PMTs) (H10770P, Hamamatsu). MATLAB script-based data acquisition software ScanImage (*Pologruto et al., 2003*) was used to acquire images. Imaging laser intensity (910 nm) was controlled by Pockels cells, with laser power at the sample plane of 10–15 mW. For glutamate uncaging experiments, to isolate NMDAR currents the following drugs were included in the ACSF bath: 1 µM TTX (Tocris, 1069), 1 µM Gabazine (Tocris, 1262), 1 µM NBQX (Tocris, 0373), 1 µM strychnine (Sigma-Aldrich, S8753), 1 µM Pyr-3 (Tocris,3753), 1 mM MNI-caged-L-glutamate. Images were acquired at ×2 magnification, sampled at 15.6 Hz continuously, 1 ms dwell time per pixel, 256 × 256. Uncaging was performed using 1 ms pulses at 725 nm power (<25 mW). Each trial was 15 s long with a 60 s ITI between trials. The mean NMDAR current for each cell was calculated across all trials that evoked a response. For each experiment we uncaged on several spots next to the

dendrite of the UBCs. For analysis of uncaging-evoked NMDAR currents, we excluded trials which did not evoke a current (using a 4 pA threshold).

For two-photon experiments imaging GRAB$_{DA2h}$ data were acquired with a 3.9 Hz sampling rate, each trial lasted ~25 s with an ITI of 150 s. A monopolar stimulating electrode was placed 100–200 µm away from cell being imaged within white matter tracts. The electrical stimulation (50 pulses, 50 Hz, 100 µs pulse width, 100–300 µA) was delivered every 150 s. About 5–10 baseline trials were acquired before the flow-in of Drd2 antagonist (1 µM L-741,626). For optogenetic experiments, the same imaging parameters as for electrical stimulation experiments were used. Instead of electrical stimulation, CoolLED pe300 (CoolLED Ltd., Andover, UK) was used to activate ChR2 (200 pulses at 39 Hz, 2ms pulses, 460 nm, 1–2 mW). For these experiments, we visually verified the expression of ChR2+ in LC axons post-hoc, using the fused mCherry fluorophore. To avoid activating TH+ Pkj cells, we excluded acute slices from animals with Pkj cells expressing mCherry.

## Pharmacology

All drugs were purchased from Sigma Aldrich (St. Louis, MO), or Tocris (Bristol, UK). SKF 81297 hydrobromide (Tocris, 1447) was applied either as puff (500 µM) or flow in (10 µM) depending on experiment type. All other drugs were bath applied: SCH-39166 hydrobromide (Tocris,2299), SCH-23390 hydrochloride (Tocris, 0925), SKF-83566 hydrobromide (Tocris, 1586), Strychnine (Sigma-Aldrich, 8753), Tetraethylammonium chloride (Sigma-Aldrich, 86614), TTX (Tocris Bioscience, 1069), (RS)-CPP (Tocris Bioscience, 0173), NBQX (Tocris Bioscience, 0373), SR 95531 hydrobromide (Tocris Bioscience, 1262), MNI-caged-L-glutamate (Tocris Biosciences 1490), L-741,626 (Tocris Bioscience, 1003), Pyr-3 (Tocris,3753), prazosin (Sigma-Aldrich, P7791), propranolol (Sigma-Aldrich, 40543), and quinpirole hydrochloride (Tocris Bioscience, 1061). For experiments requiring pharmacological agents dissolved in DMSO the concentration never exceeded 0.02% DMSO.

## Analysis and statistics

No statistical methods were used to pre-determine sample sizes, but our sample sizes are similar to those reported in previous publications. Custom MATLAB (MathWorks, Natick, MA) scripts were used for analysis of electrophysiology data. For current clamp recordings, we analyzed membrane voltage, firing rate, input resistance, and action potential half width. Membrane voltage was calculated using a median filter to exclude changes due to action potential firing. Data from cell-attached recordings were used to quantify spontaneous firing rate without perturbing the membrane. Data from voltage clamp recordings were first baseline subtracted and then analyzed for width, peak of evoked current traces, and total charge of evoked currents. For all electrophysiological recordings, reported mean values from each cell are the average of 5–10 trials. Fluorescence intensity signals from GRAB$_{DA2h}$ imaging experiments were measured in ImageJ/FIJI (*Schindelin et al., 2012*), data were then analyzed in MATLAB. All code used for analysis is available here: https://github.com/KozorovitskiyLaboratory/Canton-et-al-2022, (copy archived at swh:1:rev:3aae9c81e432920dc0be8c7af07e-c48950e8dcb9; *Canton-Josh et al., 2022*). Fluorescence signal for each trial was normalized as ΔF/F. Summary values were aggregated in GraphPad Prism (GraphPad, LaJolla, CA) for all statistical tests. All population data were tested for normality using Shapiro-Wilk normality test, D'Agostino & Pearson omnibus normality test, and KS normality test. Paired t-tests or paired Mann-Whitney U tests were used, as appropriate. For comparisons between current clamp datasets, we used the Kruskal-Wallis test with Dunn's multiple comparison post-hoc tests. In all figures, bars represent mean ± SEM. To compare EPSC/IPSC ratios and changes in Pkj cell firing rates, we used simple linear regression. Group allocation was randomized. All data analysis used batch processing with the same code and parameters. For histological analyses comparing expression of proteins or mRNA between brain regions, the same intensity and size thresholds were applied.

## Acknowledgements

This work was supported by NINDS R01NS107539, NIMH R01MH117111, Searle Scholar Award, the Beckman Young Investigator Award, and the Rita Allen Scholar Award. J.E.C. was supported as a trainee on an NIH/NINDS institutional training grant (T32NS041234) and with a Ruth Kirschstein National Research Service Award (NRSA) Individual Predoctoral Fellowship (NINDS 1F31NS120736). We thank Dr. Tiffany Schmidt and Northwestern University Biological Imaging Facility for the use of

their confocal microscopes, and Lindsey Butler for mouse colony management. We are grateful to Dr. Bernardo Sabatini for the gift of retrograde AAV virus, Dr. Horwitz for the gift of Pcp2 promoter ChR2 AAV, Dr. Yulong Li for sharing the GRAB$_{DA2h}$ virus, and Dr. Raj Awatramani for the Th-Flpo mouse line which we used to isolate Tyrosine hydroxylase+ neurons. We thank Dr. Michael Priest and Samuel Minkowicz for feedback on this manuscript.

## Additional information

### Funding

| Funder | Grant reference number | Author |
|---|---|---|
| National Institute of Neurological Disorders and Stroke | R01NS107539 | Yevgenia Kozorovitskiy |
| National Institute of Mental Health | R01MH117111 | Yevgenia Kozorovitskiy |
| Rita Allen Foundation | Rita Allen Scholar Award | Yevgenia Kozorovitskiy |
| Kinship Foundation | Searle Scholar Award | Yevgenia Kozorovitskiy |
| National Institute of Neurological Disorders and Stroke | T32NS041234 | Jose Ernesto Canton-Josh |
| National Institute of Neurological Disorders and Stroke | F31NS120736 | Jose Ernesto Canton-Josh |

The funders had no role in study design, data collection and interpretation, or the decision to submit the work for publication.

### Author contributions

Jose Ernesto Canton-Josh, Conceptualization, Data curation, Formal analysis, Funding acquisition, Investigation, Methodology, Resources, Software, Validation, Visualization, Writing - original draft, Writing - review and editing; Joanna Qin, Joseph Salvo, Formal analysis, Investigation, Writing - review and editing; Yevgenia Kozorovitskiy, Conceptualization, Data curation, Funding acquisition, Methodology, Project administration, Resources, Supervision, Writing - original draft, Writing - review and editing

### Author ORCIDs

Jose Ernesto Canton-Josh ⓘ http://orcid.org/0000-0002-1965-1079
Yevgenia Kozorovitskiy ⓘ http://orcid.org/0000-0002-3710-1484

### Ethics

All experiments were performed under the guidelines set by Northwestern University Institutional Animal Care and Use Committee (approved protocol IS00002086) .

### Decision letter and Author response

Decision letter https://doi.org/10.7554/eLife.76912.sa1
Author response https://doi.org/10.7554/eLife.76912.sa2

## Additional files

### Supplementary files
• Transparent reporting form

### Data availability

All data generated or analyzed during this study are included in the manuscript and supporting files. Source data files are provided for each figure.

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
