## [Editor Report]

This paper uses an impressive battery of anatomical and functional approaches to establish that a subpopulation of unipolar brush cells (UBCs) in the vestibulocerebellum express Dr1a dopamine receptors and that these cells are a target of locus-coeruleus-mediated dopamine release. It also demonstrates a recurrent microcircuit between UBCs and Purkinje cells, and highlights possible consequences of dopaminergic modulation of this microcircuit on cerebellar output. It will be interesting to see future in vivo studies examine the functional impact of the findings described here, which extend our understanding of neuromodulation in the cerebellum.

---

## [Decision Letter]

**Decision letter after peer review:**

[Editors’ note: the authors submitted for reconsideration following the decision after peer review. What follows is the decision letter after the first round of review.]

Thank you for submitting your work entitled "Dopaminergic regulation of vestibulo-cerebellar circuits through unipolar brush cells" for consideration by *eLife*. Your article has been reviewed by 3 peer reviewers, and the evaluation has been overseen by a Reviewing Editor and a Senior Editor. The reviewers have opted to remain anonymous.

Comments to the Authors:

Although the reviewers found the work to be interesting and well done, there was consensus that the circuit-level effect of the dopamine/UBC pathway had not been adequately established. As this was seen as the most interesting part of the paper, the conclusion was reached that the new data required falls outside the range expected for a revise decision at *eLife*. If you are able to more fully develop this aspect of the study, we would be open to considering a resubmission in the future.*Reviewer #1:*

This manuscript makes two valuable contributions to the field of cerebellar circuitry. First, although it has previously been shown that cerebellar Purkinje cells have a modest feedback inhibitory loop onto granule cells, this manuscript shows that unipolar brush cells are also targets of this feedback inhibition. These data are interesting in their own right and also imply that the entire view of Purkinje cells as a mostly feedforward element must be deeply reconsidered. Although UBCs are mostly prevalent in vestibular regions of the cerebellum in mouse, they are found more broadly in later vertebrates, so this result may be widely important.

Second, the authors find that a subset of UBCs receive dopaminergic inputs likely from the locus coeruleus. Again, although there has been prior research on this general topic, this study shows the specificity of this input and is therefore an important contribution to the literature.

Two elements of the results were weaker or confusing. First, the authors claim that UBC activation leads to pauses in Purkinje cell activity (Figure 3). However, their own data do not particularly support this claim; instead, most cells are unaffected by optogenetic activation of UBCs in slice. Selective analysis of a subset of cells that are post-hoc identified as "pausers" is not persuasive. This result does not impact the interpretation of other elements of the manuscript, and likely is a consequence of experimental design (unreliable axonal projections in slice preparation). The claims regarding "pausing" should be removed, and if the authors really want to show "pauser" PC responses (Figure 3G) then they should also show "burster" responses on the same graph.

Second, the evidence for dopaminergic inputs to UBCs is somewhat confusing. The electrical stimulation experiment was carried out in the absence of any synaptic blockers. Presumably, the authors want NMDA-R to remain intact as possible "carriers" of the signal, but would it be reasonable to block ionotropic glutamate receptors in this experiment in order to minimize the chance that these are polysynaptic effects? Alternatively, would it be feasible to block synaptic inputs (except for dopamine) and more clearly isolate the presumed effects on potassium leak channels, given the changes in input resistance elicited by SKF? The authors should explain the rationale and interpretation of these experiments more clearly.

Relatedly, a point that is still important to address is the presence of Dat in the LRN afferents to the cerebellum. While the authors state that there is no prior evidence that LRN can release dopamine, the fact that there are labeled Dat+ terminals from the LRN in the vicinity of these UBCs seems significant. Can the authors identify whether these LRN neurons contain Vmat2, as they did with Purkinje cells, to explore the possibility that they are a contributing source of dopamine? The data presented are fairly strong evidence that LC is a source of dopamine, so even if LRN is positive for Vmat2 that would not invalidate these results, but the authors claim it is not a possible source simply on the absence of evidence, which seems to need validation.

– Line 282, the authors don't provide any quantification of the virus tracing experiments. There must be some element that could be quantified, such as relative number of PCs, Golgi cells, and vestibular nucleus neurons that express GFP.

– Figure 4F is never referred to in the main text.

– It would be helpful to see a longer baseline period during the IPSC recordings (Figure 4) to have a better sense of the spontaneous rate of IPSCs in UBCs. It was unclear whether the quantification of IPSC responses in Figure 4d was baseline subtracted or not.

– Line 450-451, sentence fragment.

– Line 499-500, I understood the Balmer paper as showing that the semicircular canals, not the otoliths, had a privileged line of communication to ON UBCs (i.e. primary afferents).

– The Methods don't seem to include any measurement or reporting of the junction potential, which makes it difficult to interpret the values given for voltage clamp. Are these values already corrected for junction potential, or not, and how large is it?

*Reviewer #2:*

This study examined a variety of topics surrounding the presence of DRD1+ UBCs in vestibular cerebellum. A wide variety of imaging, electrophysiological, optogenetic, and molecular genetic tools were employed. The primary conclusions are (1) that DrD1 positive UBCs are weakly excited by dopamine receptor agonists, both through effects on intrinsic membrane properties and enhancement of NMDAR, (2) that the source of dopamine to these neurons may be locus coeruleus, (3) that DrD1+ UBCs briefly pause firing in downstream Purkinje cells, (4) that Purkinje cells make direct inhibitory contacts onto these Drd+ UBCs. However, the study's two branches, dopaminergic transmission and microcircuitry of cerebellar cortex, do not seem to hold together as a single work. For example, perhaps UBC-Purkinje cells interactions are common to all UBC subtypes. Related to this last point, the authors could have done more to place Drd1+ UBCs in the context of the many studies of UBC subtypes, for example the so-called ON cells characterized by mGluR1 and GRP expression. Without that characterization, the presentation is incomplete. A number of pharmacological effects seem rather weak and raises the question of how impactful the study will be. This includes both the effect of the dopamine receptor agonist and the effect of UBCs on Purkinje cells firing.

The choice to study NMDAR was based on published studies of prefrontal cortex. However, there is not a lot of evidence that NMDAR mediate much functional transmission at the UBC synapse, and there is more evidence that AMPAR and mGluRs do. This concern is compounded by the small effects that were found on NMDAR.

In the Discussion, the authors talk about on beam and off beam granule cell axons and how there could be differential excitation and inhibition effects of UBCs on Purkinje cells as a result. This is quite interesting, but invalidates the experimental design of Figure 3 where all the Drd1+ cells are activated at once, and a mixed, weak effect is seen in the Purkinje cell.

Another concern is that rabies transsynaptic labeling was used to explore synaptic partners of the UBCs, yet the authors never mentioned whether the method labeled locus coeruleus neurons, which are claimed to be the source of dopamine to UBCs.

The authors should have done whole cell recordings from UBCs during the LC optogenetic stimulation used in Figure 6 in addition to their GRABda measurements. Do these fibers co-release glutamate? They refer to 'DA co-release' but don't show that anything other than that DA is released. Moreover, whole cells recordings could test whether the release of DA measured by GRABda has any effect on the Drd1 UBCs. So this was a significant omission.

The authors could also colabel for mGluR1 in their drd1-td mice and thereby place these drd1 cells in the context of previous studies of UBC 'subtypes'. I would suggest splitting this paper into a drd1 study and a UBC circuitry study, although it is likely that more experiments will be needed for each.

*Reviewer #3:*

In this study, Canton-Josh et al. use an impressive battery of anatomical and functional approaches to establish that a subpopulation of unipolar brush cells (UBCs) in the vestibulocerebellum express Dr1a dopamine receptors and that these cells are a target of locus-coeruleus-mediated dopamine release. This finding is important because neuro-modulatory pathways in the cerebellum are relatively undefined and the source of dopamine signaling in the cerebellum is somewhat controversial (i.e., Purkinje cells [PCs] have long been suspected of being dopaminergic but the authors nicely show that these cells do not have the requisite molecular machinery to package and release dopamine). That said, the manuscript falls short in the authors' attempt to establish a circuit-level effect of this dopamine signaling pathway. Specifically, although the authors show that UBCs and Purkinje cells (PCs) form a recurrent microcircuit, they do not directly test the exact effect of dopamine signaling on this microcircuit. Without this critical piece of information, it is difficult to anticipate the physiological consequence LC dopamine signaling on cerebellar function.

Although the authors nicely set up a potential microcircuit between UBCs and PCs, they do not directly test how dopamine signaling actually alters recurrent activity within this microcircuit. If UBCs spike more in the presence of dopamine, what is the effect on PC throughput and how does inhibitory PC feedback onto UBCs temper this response?

The authors show that stimulating Drd1-expressing UBCs affects the firing of some PCs (a few cells increase their spiking, some cells decrease their spiking, but most PCs are unresponsive). What is the mechanistic explanation for this disparate result (the pathway seems unresolved)?

The authors' connectivity mapping experiments seems incomplete. For example, does optogenetic Purkinje cell activation drive inhibitory signaling onto non-Dr1a-expressing UBCs? Also, only half of Dr1a-expressing UBCs show an inhibitory response to PC stimulation. Are the authors implying a segregation of PC inhibitory connectivity onto UBC subtypes (including both Dr1a and non-Dr1a-expressing cells)?

The authors nicely use optogenetic LC stimulation to show evoked dopamine release onto Dr1a-expressing UBCs. However, the authors do not use any functional tests to examine for evoked dopamine release driven from SNc/VTA or Purkinje cell optogenetic stimulation (they only use anatomical mapping and/or molecular profiling to discount the possibility of release from these other potential sources).

Co-release of dopamine and NE from LC fibers is inferred though not directly shown experimentally. Relatedly, do Dr1a-positive UBCs also express adrenoreceptors?

Were the functional recordings all performed in the vesitibulo-cerebellum?

[Editors’ note: further revisions were suggested prior to acceptance, as described below.]

Thank you for resubmitting your work entitled "Dopaminergic regulation of vestibulo-cerebellar circuits through unipolar brush cells" for further consideration by *eLife*. Your revised article has been evaluated by Gary Westbrook (Senior Editor) and a Reviewing Editor. The manuscript has been improved but there are some remaining issues that need to be addressed, as outlined below. The most significant concern relates to the magnitude and significance of the dopaminergic effects.

Recommendations for the authors:

While the authors have used state-of-the-art tools and extremely thorough analyses to characterize functional Drd1 expression in UBCs, the ultimate impact on cerebellar function remains unclear. There is consensus that the results are still interesting, both in terms of the new characterizations of UBCs and from the perspective of co-transmission in general. However, the limitations to the conclusions that can be made regarding the impact on cerebellar function need to be stated and discussed more explicitly. In particular, the conditions under which DA release might occur and the extent to which it could affect cerebellar function in vivo need to be clearly discussed. A discussion of the limitations of the slice preparation could also be helpful (For example, these sparse LC inputs might be weak in slice preparation, but could be stronger under particular behavioral circumstances). Please revise the manuscript to make these limitations more explicit as well as responding to the specific comments of the reviewers, below.

*Reviewer #1:*

The authors addressed my concerns raised in the first submission.

*Reviewer #2:*

In this revision, Canton-Josh and colleagues have addressed prior reviews thoroughly. The revised manuscript presents an exhaustive set of experiments evaluating dopamine's effects on a subset of UBCs and Purkinje cell feedback to that population.

I support publication of this manuscript. However, I have one minor concern which is the strength of the authors' conclusion that LC is the source of dopamine to UBCs in Lobule X. FIg. 4 shows that hardly any LC neurons were retrogradely labeled by the viral strategy; Fig. 5F shows that lobule X seems uniquely low in LC inputs, compared to the rest of the cerebellum; and Fig. 5G shows a fairly weak GRAB-da response even to 200 pulses of LC neurons. The authors have done an admirable job delineating the possible sources of dopamine here and trying to rule out other options based on their numerous labeling and in situ experiments, and I don't have any other ideas for where this dopamine input might come from. Perhaps the authors could simply acknowledge the relative sparseness here as a consideration in interpretation.

*Reviewer #3:*

This paper proposes an interesting link between Purkinje cells and UBCs that may be selectively modulated by dopamine coming from locus coeruleus projections. While a tremendous number of state-of-the-art experiments are shown, I remain concerned about the magnitude of the actions of synaptically released DA onto UBCs and the impact downstream to Purkinje cells.

For example, Figure 2 describes the key point that DA accelerates firing in UBCs. The legend of the figure says that the agonist SKF81297 was applied with a 300 ms puff. The effect was a 110% increase over baseline, though with enormous scatter and a total increase of only 2.4 Hz. However, if one looks in the Methods, one finds the following description of how this experiment was performed: "The baseline period lasted ~7 min (20 trials) before we applied SKF81297. To activate Drd1 receptors we puff applied SKF81297 (500 µM) using a 300 ms long PicoSpritzer puff (Parker Hannifin, Hollis, NH). The drug was applied once per trial (20 s ITI) for a 10-minute period (30 trials). We continued recording after drug application for a 10-minute (30 trials) washout period." Thus, we are not dealing with an acute effect of dopamine, as implied perhaps by saying it was a 300 ms puff, but rather a gradual change over 10 minutes of repeated applications. It is not uncommon that in recording spontaneous activity from small, high resistance cells, spike rates could change over 10 minutes. Moreover, the rate change upon washout of SKF is not shown.

Then the authors repeat this in the presence of SKF plus 3 DrD1 antagonists all mixed together. Why 3 of the same kind of drug? Line 153 says that this manipulation blocked the effect of SKF on Vm and Rin, but the figures 2F,G shows that the differences were not significant ('ns').

The authors responded to my concern about the small size, and therefore low impact, of NMDAR responses by citing vanDorp and de Zeeuw 2014 as support for NMDAR function in UBCs. That must be an error, as that paper showed simply that a mixture of AMPA and NMDA blockers added together eliminated synaptic responses: "In response to five pulses at 200 Hz, a slow EPSC was observed that rose to a maximal amplitude of ~20 pA in ~200 ms in these two cells. Bath coapplication of NMDAR antagonist D-APV and AMPAR antagonist CNQX abolished the fast response and the subsequent slow wave, confirming their synaptic nature (gray traces)." Thus nothing can be concluded from that experiment about the receptor subtypes mediating the different parts of the EPSC.

Moreover, the NMDAR experiments remain concerning. In Figure 2L-O why is the NMDAR current smaller in response to uncaging than synaptic stimulation? It should be the other way around, since the exogenous glutamate reaches a larger area than a synapse. Also, are the same data represented in N and O? There appear to be several responses over 15 pA in SKF for O but not for panel N.

Figure 4 retrograde rabies experiment labeled just a few LC neurons, and in the figure (4C, far right) these are very hard to see convincingly. So, I feel that this is weak evidence for this projection being important physiologically.

Figure 5 GRABda showed a large response to electrical stimulation of fibers. Yet ChR2 activation of LC fibers caused a weak response, in 5/20 cells, and in less than half of the trials. The best evidence so far that LC projects to Lobe X is from the experiment in which the AAV1-FlexFRT-ChR2 virus was injected into the LC of TH-FLPO mouse and then image lobe X. Yet in Figure 5F it seems that there is no label in lobe X. Fig5S1 has a higher power image of lobe X that should be moved to the main figure, yet it shows that the labeled fibers are in the molecular layer, not where the UBCs are. Additionally, we still do not know if such stimulation of LC cells would lead to an ionic current or NMDAR modulation in UBCs. ChR2 stim of ChR2 expressing LC axons barely increased firing (increase of 1.2 Hz).

Figure 6 – Activation of all of the DrD1 UBCs at the same time with ChR2 can sometimes increase or decrease the firing rate of Purkinje cells, but usually does nothing.

Figure 7 – Activating Purkinje cells causes unreliable IPSCs in UBCs with long latency. In 11/24 cells they saw IPSCs. In 60% of trials of 100 light pulses there was at least one IPSC.

---

## [Author Response]

[Editors’ note: the authors resubmitted a revised version of the paper for consideration. What follows is the authors’ response to the first round of review.]

Reviewer #1:This manuscript makes two valuable contributions to the field of cerebellar circuitry. First, although it has previously been shown that cerebellar Purkinje cells have a modest feedback inhibitory loop onto granule cells, this manuscript shows that unipolar brush cells are also targets of this feedback inhibition. These data are interesting in their own right and also imply that the entire view of Purkinje cells as a mostly feedforward element must be deeply reconsidered. Although UBCs are mostly prevalent in vestibular regions of the cerebellum in mouse, they are found more broadly in later vertebrates, so this result may be widely important.Second, the authors find that a subset of UBCs receive dopaminergic inputs likely from the locus coeruleus. Again, although there has been prior research on this general topic, this study shows the specificity of this input and is therefore an important contribution to the literature.Two elements of the results were weaker or confusing. First, the authors claim that UBC activation leads to pauses in Purkinje cell activity (Figure 3). However, their own data do not particularly support this claim; instead, most cells are unaffected by optogenetic activation of UBCs in slice. Selective analysis of a subset of cells that are post-hoc identified as "pausers" is not persuasive. This result does not impact the interpretation of other elements of the manuscript, and likely is a consequence of experimental design (unreliable axonal projections in slice preparation). The claims regarding "pausing" should be removed, and if the authors really want to show "pauser" PC responses (Figure 3G) then they should also show "burster" responses on the same graph.

1.1. We have moderated the claims and clarified the description of potential pausing behavior in Pkj cells. As requested, we have added the average traces for cells showing a bursting response in Figure 6G. We have also changed Figure 6A (former Figure 3A) to include a schematic for the connectivity patterns that can generate pausing and bursting behavior.

The goal of this experiment was to take advantage of our transgenic line which labeled UBCs and measure how an increase in firing of UBCs, in this case optogenetic activation, could alter the Purkinje cells, the output of the cerebellar cortex. The Reviewer is correct in pointing out the limitations of probing multi-synaptic circuit mechanisms in the acute slice, which is acknowledged in the new Discussion section (Lines 553-557). There are several reasons listed below for why simultaneous activation of UBCs in slice may offer insights into UBC function in vivo.

– Drd1+ UBCs are mostly mGluR1+ (Figure 1D), classifying them as ON UBCs, known to fire long bouts of action potentials in response to glutamate release onto their brush (Borges-Merjane and Trussell, 2015).

– All UBCs contact several postsynaptic partners, including other UBCs (Nunzi et al., 2001). Thus, numerous UBCs could be activated within a short time period in vivo due to post-synaptic partner divergence and feed-forward excitation. We have expanded the Discussion section outlining the need for future in vivo experiments that can precisely estimate the degree of inter-connectivity of UBCs in the granular layer circuits (Lines 563-573).

– Concurrent activation of multiple UBCs leads to large areas of the granular layer being activated, which favors the activation of inhibitory inputs from molecular layer interneurons onto Purkinje cells (Dizon and Khodakhah, 2011) (Lines 558-565).

The proposed circuit mechanism is likely not the only way for UBCs to alter the activity of Purkinje cells, but a reasonable possibility given their location in the granular layer circuit. Widespread activation of many mossy fiber inputs would have a similar effect. Whether the preference for Pkj cell ‘pausing’ we described here is a unique feature of circuits which include UBCs will depend on many factors including the in vivo activity patterns of mossy fibers in the cerebellum of behaving animals, as is now detailed in the Discussion section (Lines 566-573).

Second, the evidence for dopaminergic inputs to UBCs is somewhat confusing. The electrical stimulation experiment was carried out in the absence of any synaptic blockers. Presumably, the authors want NMDA-R to remain intact as possible "carriers" of the signal, but would it be reasonable to block ionotropic glutamate receptors in this experiment in order to minimize the chance that these are polysynaptic effects?

1.2. Experiments measuring NMDA receptor currents in response to electrical stimulation were done in the presence of synaptic blockers for AMPA, GABA, and glycine channels (Lines 164-166). Additional glutamate uncaging experiments with NMDAR current recordings, which bypass presynaptic terminals altogether, help rule out any polysynaptic effects. We have clarified this in the text (Lines 168-175, 510-512).

Alternatively, would it be feasible to block synaptic inputs (except for dopamine) and more clearly isolate the presumed effects on potassium leak channels, given the changes in input resistance elicited by SKF? The authors should explain the rationale and interpretation of these experiments more clearly.

1.3. The activation of GPCRs leads to complex downstream effects. Even in neuronal classes studied across dozens of labs for many decades (*e.g.,* striatal spiny projection neurons) the mechanisms downstream of Drd1 receptor activation are still being explored (Chuhma et al., 2018; Jones-Tabah et al., 2021; Lahiri and Bevan, 2020). We agree with the Reviewer that an experiment measuring potassium leak currents in UBCs would be worthwhile, but our goal in this paper was to validate functional Drd1 expression in these cells. We believe elucidating the many effects of Drd1 activation in UBCs would be ideal for future follow-up work in another paper.

Relatedly, a point that is still important to address is the presence of Dat in the LRN afferents to the cerebellum. While the authors state that there is no prior evidence that LRN can release dopamine, the fact that there are labeled Dat+ terminals from the LRN in the vicinity of these UBCs seems significant. Can the authors identify whether these LRN neurons contain Vmat2, as they did with Purkinje cells, to explore the possibility that they are a contributing source of dopamine? The data presented are fairly strong evidence that LC is a source of dopamine, so even if LRN is positive for Vmat2 that would not invalidate these results, but the authors claim it is not a possible source simply on the absence of evidence, which seems to need validation.

1.4. To address the concern regarding expression of mRNA necessary for synthesis and release of DA in the LRN, we performed additional in situ hybridization experiments. We used a probe for Vglut2 mRNA as a marker for LRN neurons (Hisano et al., 2002; Li et al., 2020) and then co-labeled with probes for tyrosine hydroxylase and Vmat2 mRNA. This provides further evidence that LRN neurons are unable to synthesize or package dopamine (Figure 3-S1B-D; n=3 mice, *Th*,19/135 LRN cells, *Vmat2* 2/135 LRN cells). Notably, there is a known tyrosine hydroxylase (TH)+ population adjacent to the LRN (Bucci et al., 2017), but we found very few co-labeled LRN cells with TH labeling restricted to a nearby region that has no detectable cerebellar projections.

To clarify, we have no evidence of Dat+ fibers in the cerebellum, highlighting a likely transient developmental expression of Dat in the LRN, since labeling was observed only in the reporter cross but not confirmed by adult viral injections (Author response image 1 1 for viral expression; Figure 3B-C for transgenic reporter cross). Previous studies have reported similar developmental Cre expression in other mouse lines (Harris et al., 2014; Song and Palmiter, 2018).

**Author response image 1. sa2fig1:** Dat-Cre mouse injected with a flexed reporter virus into the lateral reticular nucleus shows no expression.

– Line 282, the authors don't provide any quantification of the virus tracing experiments. There must be some element that could be quantified, such as relative number of PCs, Golgi cells, and vestibular nucleus neurons that express GFP.

1.5. To maximize starter population targeting for optimal quantification, we carried out a new set of experiments, targeting two locations along the anterior-posterior axis. We attained sufficient depth and consistency in the number of presynaptic cells and starter cells, with new quantified results (Figure 4, Lines 276-286).

– Figure 4F is never referred to in the main text.

1.6. Thank you for pointing out this omission, which has now been corrected (Line 451).

– It would be helpful to see a longer baseline period during the IPSC recordings (Figure 4) to have a better sense of the spontaneous rate of IPSCs in UBCs. It was unclear whether the quantification of IPSC responses in Figure 4d was baseline subtracted or not.

1.7. We included a new supplementary figure with longer baseline periods (Figure 7-S1). All IPSC trace recordings were baseline subtracted. The experiments measuring inhibitory postsynaptic currents (IPSCs) in UBCs were performed at room temperature to reduce spontaneous events in the slice. Additional details have been added to the methods section (Lines 856-858).

– Line 450-451, sentence fragment.

1.8. Corrected, thank you (Lines 560-561).

– Line 499-500, I understood the Balmer paper as showing that the semicircular canals, not the otoliths, had a privileged line of communication to ON UBCs (i.e. primary afferents).

1.9. Corrected, thank you (Lines 69, 276, 597).

– The Methods don't seem to include any measurement or reporting of the junction potential, which makes it difficult to interpret the values given for voltage clamp. Are these values already corrected for junction potential, or not, and how large is it?

1.10. For all whole-cell recordings, junction potential was compensated. Internal solution specific values were added to the methods section (Lines 731, 753).

Reviewer #2:This study examined a variety of topics surrounding the presence of DRD1+ UBCs in vestibular cerebellum. A wide variety of imaging, electrophysiological, optogenetic, and molecular genetic tools were employed. The primary conclusions are (1) that DrD1 positive UBCs are weakly excited by dopamine receptor agonists, both through effects on intrinsic membrane properties and enhancement of NMDAR, (2) that the source of dopamine to these neurons may be locus coeruleus, (3) that DrD1+ UBCs briefly pause firing in downstream Purkinje cells, (4) that Purkinje cells make direct inhibitory contacts onto these Drd+ UBCs. However, the study's two branches, dopaminergic transmission and microcircuitry of cerebellar cortex, do not seem to hold together as a single work.

2.1. We agree that these datasets could in principle be expanded on and separated into two different papers, despite the single focus of Drd1+ UBCs. Given time constraints and experimental limitations we combined them into this submission, because the datasets are strongly complementary. First, we characterize a new potential subclass of Drd1R expressing UBCs, identified the source of DA, and found evidence of DA release. The second portion of the paper begins the work of examining the potential functions and connectivity of these specific Drd1+ UBCs within their circuit context. Together, the complementary lines of work provide a foundational body of insights for other scientists to use and build on. In addition to adding new data, we have rearranged the Results section to highlight the interrelated goals of this work.

For example, perhaps UBC-Purkinje cells interactions are common to all UBC subtypes. Related to this last point, the authors could have done more to place Drd1+ UBCs in the context of the many studies of UBC subtypes, for example the so-called ON cells characterized by mGluR1 and GRP expression. Without that characterization, the presentation is incomplete.

2.2. By using immunohistochemistry in Figure 1, we found that the majority of Drd1+ UBCs express mGluR1 but not calretinin (CR). We did not test for expression of GRP; however, previous research indicates that GRP overlaps heavily with MGluR1 expression (Kim et al., 2012).

A number of pharmacological effects seem rather weak and raises the question of how impactful the study will be. This includes both the effect of the dopamine receptor agonist and the effect of UBCs on Purkinje cells firing.The choice to study NMDAR was based on published studies of prefrontal cortex. However, there is not a lot of evidence that NMDAR mediate much functional transmission at the UBC synapse, and there is more evidence that AMPAR and mGluRs do. This concern is compounded by the small effects that were found on NMDAR.

2.3. Previous research on NMDAR currents in UBCs show they are involved in the long depolarizing response of ON UBCs to glutamate release (van Dorp and De Zeeuw, 2014). Additionally, Calcium-permeable NMDA channels could have long term impact on UBC activity and plasticity beyond a single synaptic event (Chu et al., 2015; Pugh and Raman, 2006). Given these data, the ~15% increase in NMDAR currents we have observed is likely biologically meaningful (Lines 161-166).

In the Discussion, the authors talk about on beam and off beam granule cell axons and how there could be differential excitation and inhibition effects of UBCs on Purkinje cells as a result. This is quite interesting, but invalidates the experimental design of Figure 3 where all the Drd1+ cells are activated at once, and a mixed, weak effect is seen in the Purkinje cell.

2.4. The majority of Pkj cells (18/28) did not alter their firing rates. This observation is likely due to the strong tonic firing rate downstream of resurgent sodium currents in Pkj cells, and due to damage of axons limiting multisynaptic signaling in the acute slice preparation. Nevertheless, the Pkj cells that paused in response to UBC activation were more prevalent than those that increased their firing rate (Figure 6F). Please also see response 1.1.

Another concern is that rabies transsynaptic labeling was used to explore synaptic partners of the UBCs, yet the authors never mentioned whether the method labeled locus coeruleus neurons, which are claimed to be the source of dopamine to UBCs.

2.5. Please see response 1.5, where we provide new quantified evidence of presynaptic rabies labeling of LC neurons (Figure 4) (Lines 272-286).

The authors should have done whole cell recordings from UBCs during the LC optogenetic stimulation used in Figure 6 in addition to their GRABda measurements. Do these fibers co-release glutamate? They refer to 'DA co-release' but don't show that anything other than that DA is released. Moreover, whole cells recordings could test whether the release of DA measured by GRABda has any effect on the Drd1 UBCs. So this was a significant omission.

2.6. Several Reviewers suggested directly testing whether activation of locus coeruleus fibers could recapitulate our electrophysiological findings in Drd1+ UBCs using selective Drd1 agonist. To accomplish this, we recorded firing rates in UBCs in cell-attached mode and selectively optogenetically activated LC fibers. To isolate the effects of putative DA release from LC fibers we blocked both α and β adrenergic signaling in the acute slice. Since LC neurons can release glutamate (Fung et al., 1994; Yang et al., 2021), we blocked AMPA/NMDA channels, and TRPC3 channels which open in response to mGluR1 activation. In the optogenetic experiments we found a significant increase in firing rate across the population of UBCs. We performed additional experiments in the presence of Drd1 antagonist in the bath and did not find observe firing rate changes due to LC stimulation (Figure 5) (338-359).

The authors could also colabel for mGluR1 in their drd1-td mice and thereby place these drd1 cells in the context of previous studies of UBC 'subtypes'. I would suggest splitting this paper into a drd1 study and a UBC circuitry study, although it is likely that more experiments will be needed for each.

2.7. Given the relatively understudied circuits of vestibulocerebellar UBCs and the significance of genetic traction over them via the Drd1 receptor Cre line, we prefer to maintain this study as a single coherent unit dedicated to understanding the function and neuromodulation of these UBCs.

Reviewer #3:In this study, Canton-Josh et al. use an impressive battery of anatomical and functional approaches to establish that a subpopulation of unipolar brush cells (UBCs) in the vestibulocerebellum express Dr1a dopamine receptors and that these cells are a target of locus-coeruleus-mediated dopamine release. This finding is important because neuro-modulatory pathways in the cerebellum are relatively undefined and the source of dopamine signaling in the cerebellum is somewhat controversial (i.e., Purkinje cells [PCs] have long been suspected of being dopaminergic but the authors nicely show that these cells do not have the requisite molecular machinery to package and release dopamine). That said, the manuscript falls short in the authors' attempt to establish a circuit-level effect of this dopamine signaling pathway. Specifically, although the authors show that UBCs and Purkinje cells (PCs) form a recurrent microcircuit, they do not directly test the exact effect of dopamine signaling on this microcircuit. Without this critical piece of information, it is difficult to anticipate the physiological consequence LC dopamine signaling on cerebellar function.Although the authors nicely set up a potential microcircuit between UBCs and PCs, they do not directly test how dopamine signaling actually alters recurrent activity within this microcircuit. If UBCs spike more in the presence of dopamine, what is the effect on PC throughput and how does inhibitory PC feedback onto UBCs temper this response?

3.1. Due to the limitations of acute slice preparations and the multi-synaptic nature of this recurrent circuit the results of an experiment testing how Drd1 activation in UBCs alters Pkj feedback to UBCs would be difficult to interpret. Testing how Purkinje cells respond to long term increase of firing rate in UBCs is an important experiment but perhaps one better suited for chemogenetic manipulations of UBCs in vivo. Our experiments using optogenetic stimulation of UBCs in slice probes how an increase of UBC activity could change neuronal firing in a key cerebellar neuronal type, on short time scales.

The authors show that stimulating Drd1-expressing UBCs affects the firing of some PCs (a few cells increase their spiking, some cells decrease their spiking, but most PCs are unresponsive). What is the mechanistic explanation for this disparate result (the pathway seems unresolved)?

3.2. The sparseness of modulation in Pkj cells in response to UBC activation is likely due to experimental limitations. The acute slice preparation can only provide a lower bound estimate on potential effects of UBC modulation of Pkj cells, since severed connections limit our ability to detect these multi-synaptic effects.

Discussion has been expanded to consider this limitation (Lines 553-557).

Please also see responses 1.1 and 2.4.

The authors' connectivity mapping experiments seems incomplete. For example, does optogenetic Purkinje cell activation drive inhibitory signaling onto non-Dr1a-expressing UBCs? Also, only half of Dr1a-expressing UBCs show an inhibitory response to PC stimulation. Are the authors implying a segregation of PC inhibitory connectivity onto UBC subtypes (including both Dr1a and non-Dr1a-expressing cells)?

3.3. Here, we only recorded from UBCs labeled with tdT in the Drd1-Cre; Ai14 mice, targeting a subset of mGluR1+ UBCs (Figure 1D). The proportion of UBCs which receive direct inhibition from Pkj cells is likely higher in the intact circuit. Consistently, recent work in the lab of Dr. Regehr also found IPSCs in mGluR1+ UBCs in the acute slice (Guo et al., 2021).

The authors nicely use optogenetic LC stimulation to show evoked dopamine release onto Dr1a-expressing UBCs. However, the authors do not use any functional tests to examine for evoked dopamine release driven from SNc/VTA or Purkinje cell optogenetic stimulation (they only use anatomical mapping and/or molecular profiling to discount the possibility of release from these other potential sources).

3.4. It is difficult to completely rule out the possibility of release from the SNc/ VTA or TH^+^ Pkj cells. Yet, given the consistency of our anatomical and mRNA data (noted again below), we think that testing those hypotheses with optical dopamine sensors would not be worthwhile.

For SNc/VTA, we found no evidence of any projections to the cerebellum in our retrograde CTB labeling (Figure 3B-C), presynaptic rabies tracing (Figure 4), or attempts of anterograde labeling by injection reporter viruses into midbrain of Dat-Cre mice (Author response image 2). Additionally, another group examining reward responses in granule cells in mice found no projections from the VTA (Wagner et al., 2017). For Pkj cells, the lack of *Ddc* and *Vmat2* expression curtail the possibility of DA release. Previous work also suggests that tyrosine hydroxylase expressed in Pkj cells maintains a non-phosphorylated state, which suggests that it cannot function to produce L-DOPA in Pkj cells (Lee et al., 2006; Sawada et al., 2004).

**Author response image 2. sa2fig2:** Dat-Cre mouse injected with a flexed reporter virus into the ventral tegmental area shows no axonal labeling in the cerebellum.

Co-release of dopamine and NE from LC fibers is inferred though not directly shown experimentally. Relatedly, do Dr1a-positive UBCs also express adrenoreceptors?

3.5. For the first part of this question, please see response 2.4 and the new Figure 5 (Lines 338-359). For new optogenetic experiments we have blocked α (10 µM prazosin) and β (10 µM propranolol) adrenergic receptors, although there are no published reports of α or β-adrenergic receptor expression in UBCs. We have expanded the Discussion section to consider possible effects of NE on UBC activity. We have expanded the Discussion section to consider the observed DA effects in the context of broader neuromodulatory landscape in the cerebellum that includes NE (Lines 584-590).

Were the functional recordings all performed in the vesitibulo-cerebellum?

3.6 Yes, all recordings were done in the vestibulo-cerebellar regions of the vermis. This is clarified in the text (Lines 148-149).

References

Borges-Merjane, C., and Trussell, L.O. (2015). ON and OFF unipolar brush cells transform multisensory inputs to the auditory system. Neuron *85*, 1029–1042.

Chu, H.-Y., Atherton, J.F., Wokosin, D., Surmeier, D.J., and Bevan, M.D. (2015). Heterosynaptic regulation of external globus pallidus inputs to the subthalamic nucleus by the motor cortex. Neuron *85*, 364–376.

Chuhma, N., Mingote, S., Yetnikoff, L., Kalmbach, A., Ma, T., Ztaou, S., Sienna, A.-C., Tepler, S., Poulin, J.-F., Ansorge, M., et al. (2018). Dopamine neuron glutamate cotransmission evokes a delayed excitation in lateral dorsal striatal cholinergic interneurons. *ELife 7*, 1–29.

Dizon, M.J., and Khodakhah, K. (2011). The role of interneurons in shaping Purkinje cell responses in the cerebellar cortex. J. Neurosci. *31*, 10463–10473.

van Dorp, S., and De Zeeuw, C.I. (2014). Variable timing of synaptic transmission in cerebellar unipolar brush cells. Proc. Natl. Acad. Sci. U. S. A. *111*, 5403–5408.

Fung, S.J., Reddy, V.K., Liu, R.H., Wang, Z., and Barnes, C.D. (1994). Existence of glutamate in noradrenergic locus coeruleus neurons of rodents. Brain Res. Bull. *35*, 505–512.

Guo, C., Rudolph, S., Neuwirth, M.E., and Regehr, W.G. (2021). Purkinje cell outputs selectively inhibit a subset of unipolar brush cells in the input layer of the cerebellar cortex. *ELife 10*, 1–17.

Harris, J.A., Hirokawa, K.E., Sorensen, S.A., Gu, H., Mills, M., Ng, L.L., Bohn, P., Mortrud, M., Ouellette, B., Kidney, J., et al. (2014). Anatomical characterization of Cre driver mice for neural circuit mapping and manipulation. Front. Neural Circuits *8*, 1–16.

Hisano, S., Sawada, K., Kawano, M., Kanemoto, M., Xiong, G., Mogi, K., Sakata-Haga, H., Takeda, J., Fukui,

Y., and Nogami, H. (2002). Expression of inorganic phosphate/vesicular glutamate transporters

(BNPI/VGLUT1 and DNPI/VGLUT2) in the cerebellum and precerebellar nuclei of the rat. Mol. Brain Res. *107*, 23–31.

Jones-Tabah, J., Martin, R.D., Chen, J.J., Tanny, J.C., Clark, P.B.S., and Terence E. Hébert (2021). Dopamine D1 receptor activation and cAMP/PKA signalling mediate Brd4 recruitment to chromatin to regulate gene expression in rat striatal neurons. BioRxiv.

Kawahara, H., Kawahara, Y., and Westerink, B.H.C. (2001). The noradrenaline-dopamine interaction in the rat medial prefrontal cortex studied by multi-probe microdialysis. Eur. J. Pharmacol. *418*, 177–186.

Kempadoo, K.A., Mosharov, E. V., Choi, S.J., Sulzer, D., and Kandel, E.R. (2016). Dopamine release from the locus coeruleus to the dorsal hippocampus promotes spatial learning and memory. Proc. Natl. Acad. Sci. *113*, 14835–14840.

Kim, J.A., Sekerková, G., Mugnaini, E., and Martina, M. (2012). Electrophysiological, morphological, and topological properties of two histochemically distinct subpopulations of cerebellar unipolar brush cells. Cerebellum *11*, 1012–1021.

Lahiri, A.K., and Bevan, M.D. (2020). Dopaminergic Transmission Rapidly and Persistently Enhances Excitability of D1 Receptor-Expressing Striatal Projection Neurons. Neuron *106*, 277-290.e6.

Lee, N.S., Kim, C.T., Han, S.Y., Kawk, J.H., Sawada, K., Fukui, Y., and Jeong, Y.G. (2006). The absence of phosphorylated tyrosine hydroxylase expression in the Purkinje cells of the ataxic mutant pogo mouse. J. Vet. Med. Ser. C Anat. Histol. Embryol. *35*, 178–183.

Li, Z.H., Zhang, C.K., Qiao, Y., Ge, S.N., Zhang, T., and Li, J.L. (2020). Coexpression of VGLUT1 and VGLUT2 in precerebellar neurons in the lateral reticular nucleus of the rat. Brain Res. Bull. *162*, 94–106.

Nunzi, M.G., Birnstiel, S., Bhattacharyya, B.J., Slater, N.T., and Mugnaini, E. (2001). Unipolar brush cells form a glutamatergic projection system within the mouse cerebellar cortex. J. Comp. Neurol. *434*, 329–341.

Pugh, J.R., and Raman, I.M. (2006). Potentiation of mossy fiber EPSCs in the cerebellar nuclei by NMDA receptor activation followed by postinhibitory rebound current. Neuron *51*, 113–123.

Sawada, K., Ando, M., Sakata-Haga, H., Sun, X., Jeong, Y., Hisano, S., Takeda, N., and Fukui, Y. (2004).

Abnormal expression of tyrosine hydroxylase not accompanied by phosphorylation at serine 40 in cerebellar Purkinje cells of ataxic mutant mice, rolling mouse Nagoya and dilute-lethal. Congenit. Anom. (Kyoto). *44*, 46– 50.

Song, A.J., and Palmiter, R.D. (2018). Detecting and Avoiding Problems When Using the Cre–lox System. Trends Genet. *34*, 333–340.

Takeuchi, T., Duszkiewicz, A.J., Sonneborn, A., Spooner, P.A., Yamasaki, M., Watanabe, M., Smith, C.C., Fernández, G., Deisseroth, K., Greene, R.W., et al. (2016). Locus coeruleus and dopaminergic consolidation of everyday memory. Nature *537*, 357–362.

Wagner, M.J., Hyun Kim, T., Savall, J., Schnitzer, M.J., and Luo, L. (2017). Cerebellar granule cells encode the expectation of reward. Nat. Lett. 1–18.

Yang, B., Sanches-Padilla, J., Kondapalli, J., Morison, S.L., Delpire, E., Awatramani, R., and Surmeier, D.J. (2021). Locus coeruleus anchors a trisynaptic circuit controlling fear-induced suppression of feeding. Neuron *109*, 823-838.e6.

[Editors’ note: what follows is the authors’ response to the second round of review.]

While the authors have used state-of-the-art tools and extremely thorough analyses to characterize functional Drd1 expression in UBCs, the ultimate impact on cerebellar function remains unclear. There is consensus that the results are still interesting, both in terms of the new characterizations of UBCs and from the perspective of co-transmission in general. However, the limitations to the conclusions that can be made regarding the impact on cerebellar function need to be stated and discussed more explicitly. In particular, the conditions under which DA release might occur and the extent to which it could affect cerebellar function in vivo need to be clearly discussed. A discussion of the limitations of the slice preparation could also be helpful (For example, these sparse LC inputs might be weak in slice preparation, but could be stronger under particular behavioral circumstances). Please revise the manuscript to make these limitations more explicit as well as responding to the specific comments of the reviewers, below.

Reviewers have correctly pointed to modest effect sizes and relatively sparse innervation of LC fibers as reasons to not over-interpret this data, we have added text to address this directly in the Discussion section (Lines 489-495, Lines 555-561). Additionally, we have added details to contextualize our findings with regard to the literature on channels expression in UBCs (Lines 522-527) and recurrent connectivity between the vestibulo-cerebellum and the locus coeruleus (Lines 591-604).

Reviewer #2:In this revision, Canton-Josh and colleagues have addressed prior reviews thoroughly. The revised manuscript presents an exhaustive set of experiments evaluating dopamine's effects on a subset of UBCs and Purkinje cell feedback to that population.I support publication of this manuscript. However, I have one minor concern which is the strength of the authors' conclusion that LC is the source of dopamine to UBCs in Lobule X. FIg. 4 shows that hardly any LC neurons were retrogradely labeled by the viral strategy; Fig. 5F shows that lobule X seems uniquely low in LC inputs, compared to the rest of the cerebellum; and Fig. 5G shows a fairly weak GRAB-da response even to 200 pulses of LC neurons. The authors have done an admirable job delineating the possible sources of dopamine here and trying to rule out other options based on their numerous labeling and in situ experiments, and I don't have any other ideas for where this dopamine input might come from. Perhaps the authors could simply acknowledge the relative sparseness here as a consideration in interpretation.

The reviewer is correct in stating that our anatomical data suggest a sparse projection from the LC to lobule X. It is difficult to determine whether optogenetically evoked GRAB-DA responses were modest in amplitude due to low density of LC axons, under labeling of LC neurons with precise viral targeting, or axon resection during the acute slice procedure. It is likely due to a combination of these factors. We have added a paragraph in the discussion directly addressing these points (Lines 489-495, Lines 555-561).

Reviewer #3:This paper proposes an interesting link between Purkinje cells and UBCs that may be selectively modulated by dopamine coming from locus coeruleus projections. While a tremendous number of state-of-the-art experiments are shown, I remain concerned about the magnitude of the actions of synaptically released DA onto UBCs and the impact downstream to Purkinje cells.For example, Figure 2 describes the key point that DA accelerates firing in UBCs. The legend of the figure says that the agonist SKF81297 was applied with a 300 ms puff. The effect was a 110% increase over baseline, though with enormous scatter and a total increase of only 2.4 Hz. However, if one looks in the Methods, one finds the following description of how this experiment was performed: "The baseline period lasted ~7 min (20 trials) before we applied SKF81297. To activate Drd1 receptors we puff applied SKF81297 (500 µM) using a 300 ms long PicoSpritzer puff (Parker Hannifin, Hollis, NH). The drug was applied once per trial (20 s ITI) for a 10-minute period (30 trials). We continued recording after drug application for a 10-minute (30 trials) washout period." Thus, we are not dealing with an acute effect of dopamine, as implied perhaps by saying it was a 300 ms puff, but rather a gradual change over 10 minutes of repeated applications. It is not uncommon that in recording spontaneous activity from small, high resistance cells, spike rates could change over 10 minutes. Moreover, the rate change upon washout of SKF is not shown.

The reviewer is correct that effect change occurred over the timescale of minutes. We have included a new panel in Figure 2 supplement 1 that shows the time course of SKF81297 application. We have edited the result section to include details from methods section for clarity (Lines 145-146).

Then the authors repeat this in the presence of SKF plus 3 DrD1 antagonists all mixed together. Why 3 of the same kind of drug?

We used several Drd1 blockers out of abundance of caution because the puff-based application of Drd1 agonist, while allowing excellent spatiotemporal control, makes it difficult to estimate precise drug concentration. To minimize the likelihood of potential competitive interactions between Drd1 agonist and antagonist, we chose to use a cocktail of antagonists of different chemical structure.

Line 153 says that this manipulation blocked the effect of SKF on Vm and Rin, but the figures 2F,G shows that the differences were not significant ('ns').

The panels in figure 2 E-G are changes (δ) in FR, Vm and Rin across groups, highlighting the significant effect of Drd1 agonist application and no changes (δ around zero) in the two control conditions. We have edited Results section for clarity (Lines 148-155).

The authors responded to my concern about the small size, and therefore low impact, of NMDAR responses by citing vanDorp and de Zeeuw 2014 as support for NMDAR function in UBCs. That must be an error, as that paper showed simply that a mixture of AMPA and NMDA blockers added together eliminated synaptic responses: "In response to five pulses at 200 Hz, a slow EPSC was observed that rose to a maximal amplitude of ~20 pA in ~200 ms in these two cells. Bath coapplication of NMDAR antagonist D-APV and AMPAR antagonist CNQX abolished the fast response and the subsequent slow wave, confirming their synaptic nature (gray traces)." Thus nothing can be concluded from that experiment about the receptor subtypes mediating the different parts of the EPSC.

Thank you for pointing out the citation error. We have now cited (Billups et al., 2002; Rossi et al., 1995), which show pharmacologically isolated NMDA currents in UBCs (Line 520).

Moreover, the NMDAR experiments remain concerning. In Figure 2L-O why is the NMDAR current smaller in response to uncaging than synaptic stimulation? It should be the other way around, since the exogenous glutamate reaches a larger area than a synapse.

NMDAR current from glutamate uncaging are expected to be smaller than from electrical stimulation. For uncaging experiments, we are using 1 ms long pulses of two-photon (725 nm) uncaging of MNI-Glutamate, supporting spatiotemporally precise stimulation. These two-photon evoked uncaging post-synaptic currents are typically smaller than events evoked by electrical stimulation of presynaptic inputs or by uncaging of glutamate using 1-photon UV excitation (Passlick and Ellis-Davies, 2018). The parameters we have used are similar to single dendritic spine uncaging protocols used to study a broad range of cell classes in prior work (Bloodgood and Sabatini, 2005; Kozorovitskiy et al., 2015; Xiao et al., 2018). Moreover, mossy fibers are known to create large complex postsynaptic sites with UBCs, poised to ‘trap’ glutamate in the synaptic cleft after presynaptic firing (Kinney et al., 1997; Rossi et al., 1995).

Also, are the same data represented in N and O? There appear to be several responses over 15 pA in SKF for O but not for panel N.

In Figure 2N each dot represents the average response to stimulation for a single cell, any response with a peak amplitude over the threshold of 4 pA were included in this average. Figure 2O is the peristimulus histogram of all responses across cells in either of the two groups (ACSF, 14 cells, 466 trials; SKF, 13 cells, 349 trials). We did not perform statistical analyses on the data in Figure 2O but included the graphs to show the distribution of responses. This has been corrected in the paper (Lines 204-205).

Figure 4 retrograde rabies experiment labeled just a few LC neurons, and in the figure (4C, far right) these are very hard to see convincingly. So, I feel that this is weak evidence for this projection being important physiologically.

It is true that a relative paucity of LC neurons which project to lobules IX/X compared to rest of cerebellar cortex, although we show that these projections exist, and they release monoamines onto UBCs. We have edited the Discussion section to acknowledge limitations and highlight directions for further investigations to assess physiological relevance (Lines 489-495, Lines 555-561).

Figure 5 GRABda showed a large response to electrical stimulation of fibers. Yet ChR2 activation of LC fibers caused a weak response, in 5/20 cells, and in less than half of the trials.

First, with electrical stimulation, we are likely activating a much larger population of LC axons than with optogenetic manipulation. This is because for optogenetic activation experiments we rely on two carefully targeted injections in a double transgenic mouse (restricting GRABDA to Drd1+ UBCs and ChR2 to TH^+^ LC neurons). This challenging high specificity design is likely to underestimate actual connectivity but has no risk of false positives.

The best evidence so far that LC projects to Lobe X is from the experiment in which the AAV1-FlexFRT-ChR2 virus was injected into the LC of TH-FLPO mouse and then image lobe X. Yet in Figure 5F it seems that there is no label in lobe X. Fig5S1 has a higher power image of lobe X that should be moved to the main figure, yet it shows that the labeled fibers are in the molecular layer, not where the UBCs are.

In Figure 5F (Lines 369-371) we have included panels from Figure 5 S1 which shows a confocal image of LC fibers in Lobule X. We found fibers both in the molecular layer and granular layer. We labeled the layers in the image dot make this clearer. We did not label LC fiber terminals so cannot claim the same number of terminal sites within the molecular and granular layer. However, we do see similar structure to the axons in both layers including en passant boutons (Figure 5 S1).

Additionally, we still do not know if such stimulation of LC cells would lead to an ionic current or NMDAR modulation in UBCs. ChR2 stim of ChR2 expressing LC axons barely increased firing (increase of 1.2 Hz).

The reviewer is correct, there is a small effect size and due to release of multiple neurotransmitters from LC axons we need to be careful with interpretation. We have expanded on this in the Discussion section (Lines 555-561).

Figure 6 – Activation of all of the DrD1 UBCs at the same time with ChR2 can sometimes increase or decrease the firing rate of Purkinje cells, but usually does nothing.

Given the well-known limitations in the acute slice preparation, we expect to undercount the impact of UBC activity on Pkj firing, especially because their high intrinsic tonic firing rate moderates the effect of both excitatory and inhibitory synaptic inputs. Nevertheless, we believe that these data represent the first description of a genetically restricted UBC activity manipulation having any impact on Pkj firing.

Figure 7 – Activating Purkinje cells causes unreliable IPSCs in UBCs with long latency. In 11/24 cells they saw IPSCs. In 60% of trials of 100 light pulses there was at least one IPSC.

In the context of a dual viral labeling experiment with selective restriction on both the putative pre- and postsynaptic side, observing light-locked physiological effects in a subset of recordings is generally not considered unreliable, because the specificity of the preparation underestimates effect magnitudes. We have reported the observed variability in the amplitude and time to peak for optogenetically evoked IPSCs. It is possible that further lengthening of viral incubation times may be helpful in revealing the full magnitude of this connection (Lines 463, 472; see also Figure 7 S1A, where current amplitude scales with incubation time, all over 22 days).

References

Aston-Jones, G., Rajkowski, J., and Cohen, J. (1999). Role of locus coeruleus in attention and behavioral flexibility. Biol. Psychiatry *46*, 1309–1320.

Billups, D., Liu, Y.-B., Birnstiel, S., and Slater, N.T. (2002). NMDA Receptor-Mediated Currents in Rat Cerebellar Granule and Unipolar Brush Cells. J. Neurophysiol. *87*, 1948–1959.

Bloodgood, B.L., and Sabatini, B.L. (2005). Neuronal activity regulates diffusion across the neck of dendritic spines. Science (80-. ). *310*, 866–869.

Hayat, H., Regev, N., Matosevich, N., Sales, A., Paredes-Rodriguez, E., Krom, A.J., Bergman, L., Li, Y., Lavigne, M., Kremer, E.J., et al. (2020). Locus coeruleus norepinephrine activity mediates sensory-evoked awakenings from sleep. Sci. Adv. *6*.

Kinney, G.A., Overstreet, L.S., and Slater, N.T. (1997). Prolonged physiological entrapment of glutamate in the synaptic cleft of cerebellar unipolar brush cells. J. Neurophysiol. *78*, 1320–1333.

Kozorovitskiy, Y., Peixoto, R., Wang, W., Saunders, A., and Sabatini, B.L. (2015). Neuromodulation of excitatory synaptogenesis in striatal development. *ELife 4*, 1–18.

Passlick, S., and Ellis-Davies, G.C.R. (2018). Comparative one- and two-photon uncaging of MNI-glutamate and MNI-kainate on hippocampal CA1 neurons. J. Neurosci. Methods *293*, 321–328.

Rossi, D.J., Alford, S., Mugnaini, E., and Slater, N.T. (1995). Properties of transmission at a giant glutamatergic synapse in cerebellum: the mossy fiber-unipolar brush cell synapse. J. Neurophysiol. *74*, 24–42.

Xiang, L., Harel, A., Gao, H.Y., Pickering, A.E., Sara, S.J., and Wiener, S.I. (2019). Behavioral correlates of activity of optogenetically identified locus coeruleus noradrenergic neurons in rats performing T-maze tasks. Sci. Rep. *9*, 1–13.

Xiao, L., Priest, M.F., and Kozorovitskiy, Y. (2018). Oxytocin functions as a spatiotemporal filter for excitatory synaptic inputs to VTA dopamine neurons. *ELife 7*, 1–26.